# Molecular insights into capsular polysaccharide secretion

Jeremi Kuklewicz[1] & Jochen Zimmer[1,2 ✉]

Capsular polysaccharides (CPSs) fortify the cell boundaries of many commensal and pathogenic bacteria[1]. Through the ABC-transporter-dependent biosynthesis pathway, CPSs are synthesized intracellularly on a lipid anchor and secreted across the cell envelope by the KpsMT ABC transporter associated with the KpsE and KpsD subunits[1,2]. Here we use structural and functional studies to uncover crucial steps of CPS secretion in Gram-negative bacteria. We show that KpsMT has broad substrate specificity and is sufficient for the translocation of CPSs across the inner bacterial membrane, and we determine the cell surface organization and localization of CPSs using super-resolution fluorescence microscopy. Cryo-electron microscopy analyses of the KpsMT–KpsE complex in six different states reveal a KpsE-encaged ABC transporter, rigid-body conformational rearrangements of KpsMT during ATP hydrolysis and recognition of a glycolipid inside a membrane-exposed electropositive canyon. In vivo CPS secretion assays underscore the functional importance of canyon-lining basic residues. Combined, our analyses suggest a molecular model of CPS secretion by ABC transporters.

Bacteria commonly display complex carbohydrates on their surfaces to fortify cell boundaries[2]. In Gram-negative bacteria, CPSs are lipid-linked high-molecular-weight biopolymers that form protective capsules, serve as adhesives or camouflage pathogens inside their hosts[1]. Accordingly, CPSs are potent virulence factors and potential targets for the development of novel antimicrobial agents[3–5].

CPSs are predominantly linear, acidic polysaccharides of several hundred sugar units. Through the ABC-transporter-dependent biosynthesis pathway, CPSs are synthesized as glycolipids on the cytosolic side of the inner-membrane, followed by export across the cell envelope[1,2]. CPS secretion is mediated by an ABC transporter containing KpsT and KpsM as its nucleotide-binding domain (NBD) and transmembrane domain, respectively[6]. The transporter partners with the periplasmic and inner-membrane-anchored KpsE subunit, a class-3 polysaccharide co-polymerase (PCP), as well as with KpsD, which is assumed to form an outer-membrane pore.

CPSs are synthesized on a phosphatidylglycerol lipid anchor that is usually extended by 5-9 Kdo (3-deoxy-D-manno-oct-2-ulosonic acid) sugars by the KpsS and KpsC enzymes[7–10] (Fig. 1a–c). The Kdo glycolipid then serves as an adaptor for CPS biosynthesis either by a single bifunctional glycosyltransferase or an ensemble of glycosyltransferases, depending on the serotype. After or concomitant to biosynthesis, CPS is recognized by KpsMT and translocated through a trans-envelope conduit that consists of KpsMT, KpsE and KpsD (Fig. 1c).

KpsMT belongs to class 5 of the ABC transporters, with structural homology to the O-antigen and teichoic acid transporters that recognize undecaprenyl-diphosphate-linked glycopolymers[11–13]. In vivo substitution experiments have shown that KpsMT has limited specificity towards the CPS structure[14–16], which suggests that the conserved Kdo glycolipid moiety mediates substrate recognition.

Here we delineate how Gram-negative CPSs are recognized and secreted across the cell envelope by combining biochemical analyses with super-resolution CPS imaging and single-particle cryo-electron microscopy (cryo-EM). First, genetic engineering of *Escherichia coli* with CPS components from *Pasteurella multocida* reveals capsule formation with CPS secretion 'hotspots'. Second, cryo-EM analyses of a complex of KpsMT and KpsE in apo and nucleotide-bound states provide detailed insights into the association of the ABC transporter with a periplasmic octameric KpsE 'cage'. Third, a substrate-bound state of the KpsMT–KpsE complex reveals how KpsM recognizes the glycolipid in an electropositive canyon and identifies rigid-body movements in the transmembrane region to accommodate the substrate. Fourth, site-directed mutagenesis and in vivo functional analyses support a model by which KpsMT binds and translocates its glycolipid substrate through conserved arginine residues at the interface of the KpsM subunits.

## Broad specificity of CPS secretion

To analyse CPS biosynthesis structurally and functionally, we reconstituted CPS formation in an acapsular off-the-shelf *E. coli* laboratory strain. *E. coli* C43 (DE3) is acapsular because it lacks the functional CPS polymerase; all other CPS genes are present[17]. Therefore, the endogenous group 1 CPS gene cluster was removed from this strain by CRISPR–Cas9- and Lambda Red-mediated recombineering (Methods) to avoid functional overlap with recombinantly expressed CPS genes, giving rise to the *E. coli* C43ΔCPS1 strain (Extended Data Fig. 1a and Supplementary Figs. 2 and 3). Next, we expressed the CPS biosynthetic and secretion components from *P. multocida* (*Pm*), a Gram-negative pathogen[18], in the *E. coli* C43ΔCPS1 strain under the

[1]Department of Molecular Physiology and Biological Physics, University of Virginia School of Medicine, Charlottesville, VA, USA. [2]Howard Hughes Medical Institute, University of Virginia, Charlottesville, VA, USA. ✉e-mail: jz3x@virginia.edu

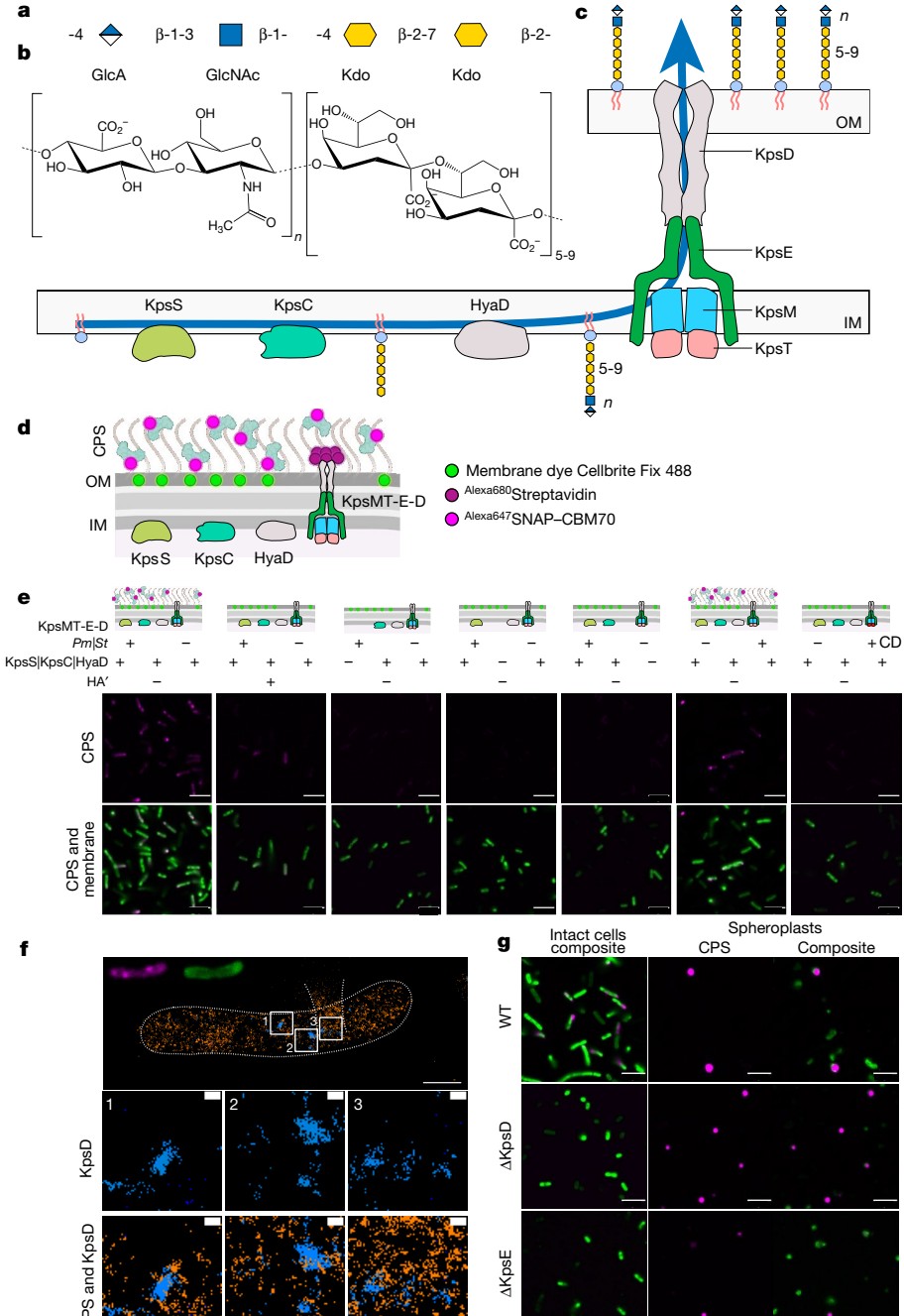

**Fig. 1 | Engineering of *E. coli* for CPS encapsulation. a**,**b**, Hyaluronan disaccharide and Kdo repeat units represented symbolically (**a**) (blue square, GlcNAc; blue and white diamond, GlcA; yellow hexagon, Kdo) and as a skeletal formula (**b**). **c**, Classical model of type-2 CPS biosynthesis with its crucial components labelled. HyaD is the *P. multocida* hyaluronan synthase. IM, inner membrane; OM, outer membrane. **d**, Diagram of the in vivo encapsulation assay and labelling strategy. KpsMT-E-D, KpsMT–KpsE–KpsD. **e**, Confocal images of *E. coli* C43ΔCPS1 expressing different CPS components as indicated at the top. Membranes were stained with Cellbrite Fix 488 and CPS with Alexa647-tagged SNAP–CBM70. HA', hyaluronidase digestion; CD, 'catalytically dead' (Walker B) KpsMT(E151Q). The *St*KpsE used in this assay contains the

engineered disulfide bridge (L77C, S138C). Scale bars, 5 µm. Parts of **c**–**e** were generated in BioRender.com. **f**, Representative dual-colour MINFLUX nanoscopy of an encapsulated cell. CPS (orange) is labelled with Alexa647-tagged SNAP–CBM70; KpsD (blue) is labelled with Alexa680-tagged streptavidin. Panels 1–3 are examples of CPS and KpsD hotspots. Scale bars, 500 nm (main image); 50 nm (panels 1–3). Volcano-like CPS scattering and the cell outline are indicated with dashed lines. **g**, Confocal images of inner-membrane-linked CPS. Spheroplasts were prepared from encapsulated (wild type (WT), top row) and acapsular but CPS-expressing cells in the absence of KpsD (ΔKpsD, middle row) or KpsE (ΔKpsE, bottom row). Membranes are stained as in **e**. Scale bars, 5 µm.

control of inducible T7 promoters (Extended Data Fig. 1b and Methods). *P. multocida* produces a hyaluronan (HA) capsule[19], consisting of linear polysaccharides of alternating units of *N*-acetylglucosamine and glucuronic acid (Fig. 1a,b). HA can be detected on the cell surface using a carbohydrate-binding module (CBM70)[20] fused to a SNAP tag charged

with an Alexa647 fluorophore (Fig. 1d–g, Extended Data Fig. 1c and Methods).

After the expression of all *Pm*CPS components, the *E. coli* C43ΔCPS1 surface can be labelled with the HA-specific probe for confocal fluorescence microscopy (Fig. 1e and Supplementary Fig. 4). Prior treatment

with a hyaluronidase substantially reduces labelling, indicating that the CBM70 probe indeed detects surface-exposed HA. Most cells exhibit distinct HA puncta that are frequently localized to the poles. A similar pattern of CPS distribution has been previously described for *E. coli* K1 and related capsules[21].

Control conditions in the absence of KpsS or KpsC, implicated in the synthesis of the Kdo-linked glycolipid anchor, do not detect extracellular HA above noise levels (Fig. 1c,e and Supplementary Fig. 4). Two additional *Pm*CPS subunits of unknown function, HyaB and HyaE, also influence encapsulation[16]. No capsule was detected in the absence of HyaB, whereas the capsule produced in the absence of HyaE has a more diffuse CPS signal with fewer HA puncta (Extended Data Fig. 1d and Supplementary Fig. 4). As expected, HA is not detected in the absence of HyaD, the bifunctional HA synthase[19] (Fig. 1c,e and Supplementary Fig. 4). Thus, although it is probably not generating a native-like capsule in its overall dimensions, the engineered system is suitable for structure–function analysis of the CPS secretion system.

As described in detail below, the CPS secretion system from the Gram-negative bacterium *Schlegelella thermodepolymerans* (*St*) is particularly suitable for structural analyses. Substituting the *Pm*CPS secretion machinery (consisting of KpsMT, KpsE and KpsD) with the *S. thermodepolymerans* counterparts in the engineered *E. coli* C43ΔCPS1 strain enables HA secretion (Fig. 1e). Under the same conditions, rendering *St*KpsT catalytically inactive by introducing the Walker B mutation (E151Q) into its NBD abolishes HA detection, showing that HA encapsulation depends on a functional *St*CPS secretion system (Fig. 1e).

High-resolution two-dimensional MINFLUX nanoscopy[22] provides insights into the capsule organization and the distribution of secretion components. Using the CBM70 probe, we imaged the capsule produced from *P. multocida* components, resolving CPS surface distributions ranging from distinct clusters to uniform coats to scattered volcano-like formations (Supplementary Figs. 5 and 6 and Supplementary Video 1). The volcano-like CPS formations are likely to correspond to CPS puncta resolved by confocal imaging when viewed from the side; that is, extending perpendicular to the imaging direction. To better estimate the protrusion of the CPS 'volcanos' extending from the cell surface, we metabolically labelled lipopolysaccharides (LPSs) with Kdo azide[23], followed by fluorophore conjugation by copper-free click chemistry (Methods). Using two-dimensional dual-colour LPS (AZ647 DBCO) and CPS ([Flux680]CBM70) MINFLUX nanoscopy[24], we estimate that the volcano-like CPS formations extend roughly 100–200 nm past the cell surface, as defined by the LPS localizations (Extended Data Fig. 1e,f and Supplementary Figs. 7–11). These dimensions might be an underestimate owing to insufficient CPS labelling and/or fixation.

Similarly, to localize CPS together with the outer-membrane secretion pore KpsD by dual-colour MINFLUX nanoscopy, an engineered C-terminal Strep tag of KpsD was detected using Alexa680-conjugated streptavidin (Extended Data Fig. 1g,h and Supplementary Fig. 12). A model of octameric KpsD predicted by AlphaFold2[25] suggests that only one or two streptavidin tetramers can bind to the KpsD complex simultaneously. Accordingly, a single KpsD octamer is expected to give rise to a fluorophore localization cloud of about 12 nm in diameter (Extended Data Fig. 1g,h).

Two- and three-dimensional MINFLUX nanoscopy detecting CPS and KpsD reveals KpsD clustering into distinct 'hotspots' ranging from around 20–50 nm in diameter (Fig. 1f, Extended Data Fig. 1i–l, Supplementary Figs. 13–20 and Supplementary Video 2). Some KpsD clusters are proximal to CPS, consistent with HA secretion sites. Although the exact number of KpsD clusters cannot be determined with high precision with the current labelling strategy (because one KpsD cluster might contain more than one secretion system), we estimate fewer than 100 clusters per cell, on the basis of their overall size and the observed KpsD localizations. KpsD clustering did not change in the absence of CPS or KpsE, indicating that KpsD oligomers probably form before

the assembly of the secretion system and/or CPS engagement (Supplementary Figs. 21 and 22). Streptavidin labelling is observed only in the presence of Strep-tagged KpsD; untagged KpsD is not detected (Supplementary Fig. 13).

## KpsMT is sufficient for CPS secretion

As discussed in detail below, structural analyses of the KpsMT–KpsE complex bound to a putative substrate molecule suggest that KpsE and KpsD are dispensable for substrate translocation across the inner membrane. We tested this hypothesis by performing in vivo CPS translocation assays in the presence or in the absence of KpsE or KpsD. In the absence of an outer-membrane pore, any translocated CPS would accumulate in the periplasm, where it can be detected after removing the outer membrane.

Expressing all but the KpsD or KpsE CPS components in *E. coli* C43ΔCPS1 did not generate any surface-exposed HA (Fig. 1g and Supplementary Fig. 23). However, in both cases, after removal of the outer membrane, several spherically shaped cells exhibited strong HA labelling, suggesting periplasmic accumulation of HA (Fig. 1g and Supplementary Fig. 23). Similar results have been obtained previously[26] for *E. coli* K5 CPS. As also observed previously[27], cells expressing all CPS components and thus displaying CPS on their surfaces are elongated, and sometimes appear as tubes. This is not the case in the absence of KpsE or KpsD. The corresponding spheroplasts are smaller in diameter, suggesting that, in the experimental conditions, CPS surface exposure affects cell division.

## KpsMT is surrounded by a KpsE cage

Structural analyses of the CPS secretion system were performed using *S. thermodepolymerans* components owing to the stability of the complex in non-denaturing detergents. The inner-membrane-associated KpsM, KpsT and KpsE subunits were co-expressed and purified for cryo-EM analyses (Methods). To stabilize the KpsE oligomer, an intermolecular disulfide bond was engineered (L77C, S138C) on the basis of an AlphaFold2-predicted KpsE dimer model, using the 'Disulfide by Design 2.0' tool[28] (Extended Data Fig. 2a,b). The obtained KpsMT–KpsE complex is catalytically active in vivo and in vitro, hydrolysing ATP at a rate of about 40 nmol min⁻¹ mg⁻¹ (Fig. 1e and Extended Data Fig. 2c). As described in detail in the next section, we determined several KpsMT–KpsE complex structures in different nucleotide-bound states (Extended Data Figs. 2d,e and 3–5 and Extended Data Table 1). The most complete KpsE assembly described hereafter was obtained with an ATP-bound KpsMT transporter.

The KpsMT transporter is surrounded by a cage of eight KpsE subunits extending about 80 Å into the periplasm (Fig. 2 and Extended Data Fig. 2f,g). The KpsMT–KpsE complex is a dimer of two KpsMT subunits, each interacting with four KpsE subunits. Starting within the membrane-embedded region, the KpsE cage can be divided into transmembrane, dome and crown regions. Most intermolecular interactions occur within the dome, which forms a tight periplasmic ring surrounding the ABC transporter (Fig. 2e–i).

KpsE is anchored to the inner membrane by N- and C-terminal transmembrane helices (Fig. 2b–e). The N-terminal helix forms the outer and the C-terminal helix the inner ring of a membrane-integrated 'carousel' (Fig. 2i). The following dome region contains a four-stranded twisted β-sheet (β1–β4), four short α-helices (α1–α4), and the N-terminal segment of helix α5, extending into the crown (Fig. 2e and Extended Data Fig. 2f). Although most of the dome region is formed from residues directly following transmembrane helix 1 (TM1; residues 34–172), the last β-strand (β4; residues 322–329) precedes the C-terminal transmembrane helix. Two loops extend from the β-sheet into the octamer lumen (L1 and L2), of which the longer L2 loop is disordered (residues 50–72) (Fig. 2b,e and Extended Data Fig. 2i). On the basis of an AlphaFold2

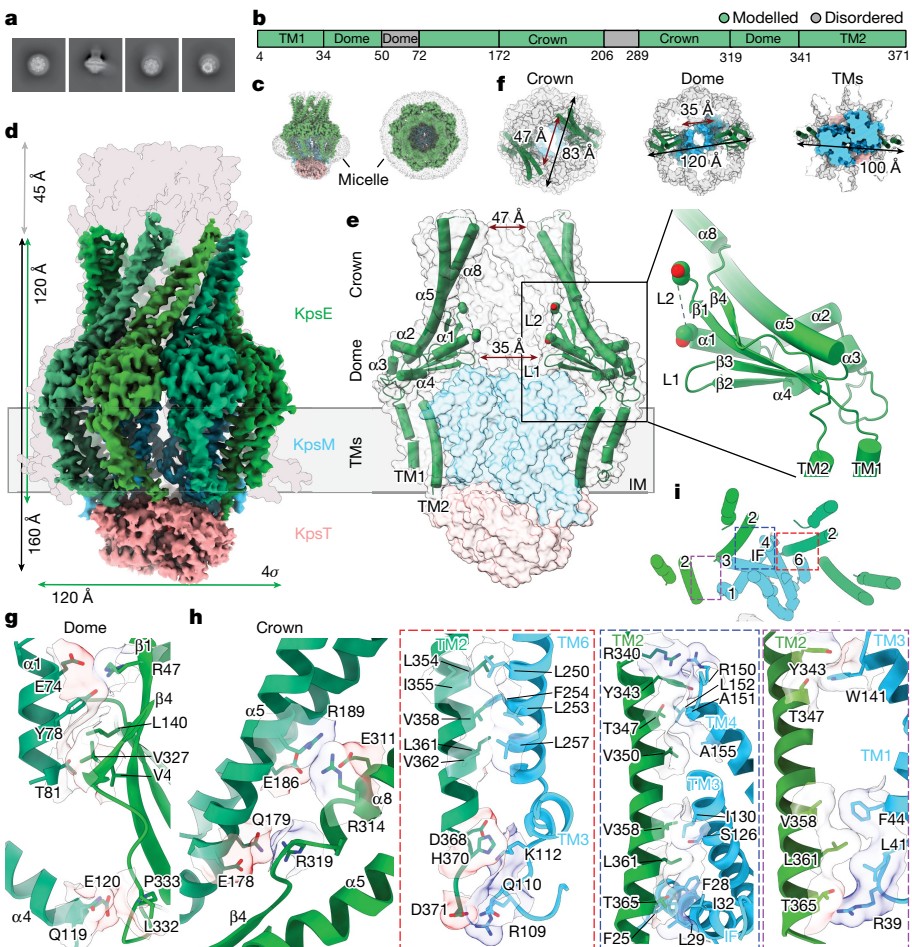

**Fig. 2 | Structure of the KpsMT–KpsE complex. a**, Representative two-dimensional (2D) class averages of KpsMT(E151Q)–KpsE. **b**, Domain organization and model completeness of KpsE. **c**, Cryo-EM map of KpsMT(E151Q)–KpsE in a lauryl maltose neopentyl glycol (LMNG) micelle shown at 4σ contour level. KpsE, green; KpsM, blue; KpsT, salmon. **d**, Map of ATP-bound KpsMT(E151Q)–KpsE overlaid with a surface representation of an AlphaFold2-predicted octameric full-length KpsE model (grey). **e**, Cartoon representation of KpsE. The last resolved residues of L2 are shown as spheres. **f**, Section views of the KpsE octamer with inner and outer diameters. **g**,**h**, Interactions between neighbouring KpsE protomers (**g**, dome; **h**, crown). **i**, Interactions between KpsE and KpsM. Viewed from the periplasm.

model, the L2 loop is likely to point away from the inner membrane into the KpsE tunnel (Extended Data Fig. 2f,g and Supplementary Fig. 12).

The crown is formed by a helical hairpin that consists of helix α5, extending from the dome, as well as helix α8 (residues 291–319). The helices are connected by a disordered region (residues 205–290) that is predicted to form another helical hairpin, which most probably interacts with KpsD (Fig. 2d,e, Extended Data Fig. 2f,g and Supplementary Fig. 12). The interior of the KpsE crown is hydrophilic (Extended Data Fig. 2h).

Starting at the cytosolic side of the membrane, the KpsE cage is about 120 Å long (Fig. 2d). Including the unresolved region extending the crown helices and on the basis of an AlphaFold2 model (Extended Data Fig. 2f,g), the assembled cage would extend by about 120 Å from the periplasmic surface of the inner membrane into the periplasm (Fig. 2d). Its interior diameter is about 35 Å and 47 Å within the dome and crown regions, respectively (Fig. 2f).

## KpsE assembly and interaction with KpsMT

Cage-stabilizing interactions between KpsE protomers occur mostly through the β-sheet of the dome (Fig. 2g). Helices α1 and α4 of a neighbouring subunit pack against the twisted β-sheet surface. Within α1, Glu74, Tyr78 and Thr81 contact the neighbouring sheet. Glu74 and Tyr78 are near Arg47 of β1, and the methyl group of Thr81 packs into a hydrophobic pocket formed by Val43 of β1, Leu140 of β3 and Val327 of β4. The N terminus of α4 contacts the loop connecting β4 with TM2 in the neighbouring subunit through Gln119 and Glu120. These residues interact with the backbone carbonyl and amide groups of Leu332 and Pro333 in the β4–TM2 connection, respectively (Fig. 2g).

The interprotomer interactions within the crown are less pronounced and are confined to the dome-proximal region. Glu178 and Gln179 of α5 are close to Arg319 at the C-terminal end of α8 in the neighbouring subunit. Similar interprotomer interactions exist between Glu186 and Arg189 of α5 and Arg314 and Glu311 of α8 (Fig. 2h).

KpsMT contacts the KpsE cage within the membrane and the periplasm (Fig. 2d–f,i). Contacts within the membrane are mostly hydrophobic, whereas charged and polar residues mediate interactions within the membrane-flanking regions. Each KpsM contacts three KpsE subunits (Fig. 2i). First, KpsM TM3 and TM6, together with the N-terminal segment of the KpsM interface helix, form a hydrophobic pocket that accommodates the end of the KpsE C-terminal transmembrane helix. Second, the C-terminal helix of the next KpsE protomer (counted anticlockwise when viewed from the periplasm) fits into a groove formed by the C-terminal region of the KpsM interface helix, the central portion of its TM3 and the N terminus of TM4 (Fig. 2i). Third, TM2 of the third KpsE subunit interacts with the C- and N-terminal segments of KpsM TM3 and TM1 on the periplasmic and cytosolic sides of the membrane, respectively.

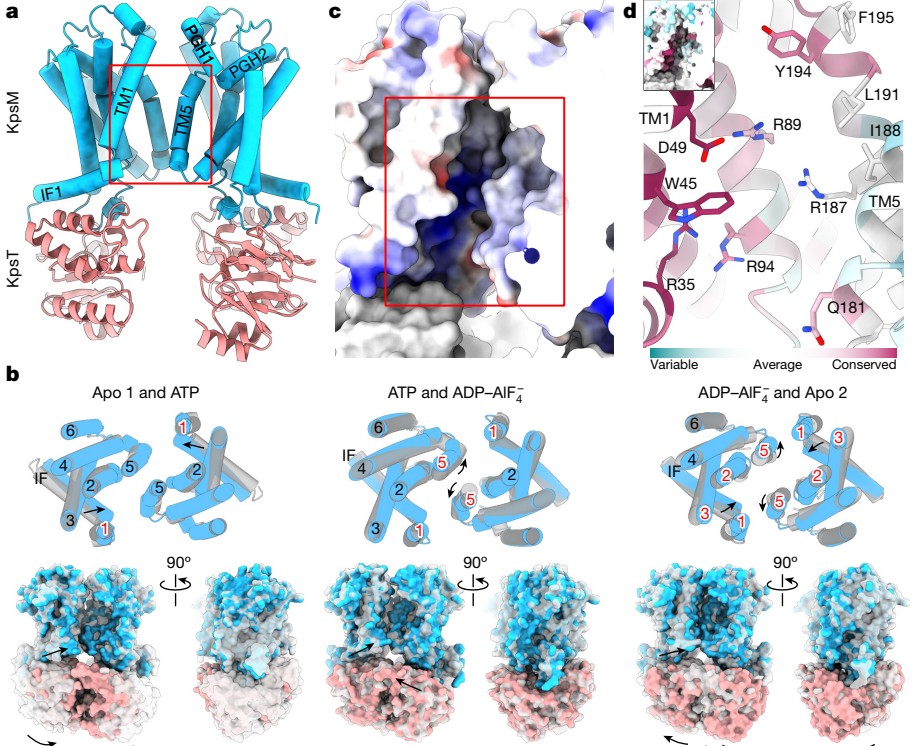

**Fig. 3 | Structure and conformational changes of the KpsMT ABC transporter.**
**a**, KpsMT Apo 1 structure. KpsM is shown as cylindrical helices in blue and KpsT as a cartoon in salmon. IF, interface helix; PGH, periplasmic gate helix. **b**, Conformational changes and rigid-body movements of KpsMT when transitioning between nucleotide-bound states (grey to coloured). Helices indicated by red font colour frame the transmembrane canyon and modulate its opening. **c**, Surface representation of the Coulombic electrostatic potential. (red, $-10$ kcal mol$^{-1}$ $e^{-1}$; blue, 10 kcal mol$^{-1}$ $e^{-1}$) of the polysaccharide canyon of KpsMT (red box). **d**, Evolutionary conservation (ConSurf[41]) of residues forming the polysaccharide canyon. Inset, surface representation of the KpsMT polysaccharide canyon coloured by sequence conservation.

## Architecture of the CPS ABC transporter

We determined cryo-EM structures of the KpsMT–KpsE complex in two nucleotide-free states (Apo 1 and Apo 2), bound to ADP–AlF$_4^-$ mimicking an ATP post-hydrolysis state, and in an ATP-complexed conformation after introducing the E151Q Walker B mutation into the active site of KpsT (Fig. 3 and Extended Data Fig. 6). In the absence of KpsE, the KpsMT transporter could be purified only after introducing the Walker B E151Q mutation, leading to co-purification with ATP. Its structure in a lipid nanodisc is nearly identical to that of the KpsMT(E151Q)–KpsE complex (Extended Data Fig. 2j) and is not further discussed.

KpsMT shows structural homology with O-antigen (WzmWzt) and teichoic acid (TarGH) ABC transporters, as well as with ABCG1 and PmtCD[12,13,29,30] (Extended Data Fig. 6c). Unlike WzmWzt and TarGH, KpsT lacks the gate helix at the interface with KpsM, which has been suggested to recognize undecaprenyldiphosphate-linked substrates[12]. The KpsMT substrate is anchored to a phosphatidylglycerol phospholipid instead[8].

KpsM starts with an N-terminal amphipathic interface helix running parallel to the cytosolic water–lipid interface, followed by six transmembrane helices (Fig. 3a,b and Extended Data Fig. 6a). In the complete transporter, the two KpsM subunits are separated from each other, with only minimal interprotomer contacts mediated by TM5, similar to other family-5 lipid ABC transporters, including ABCG1, ABCG5 and ABCG8 (refs. 29,31).

## Rigid-body movements of KpsMT

Conformational transitions between nucleotide-free, ATP and ADP–AlF$_4^-$-bound KpsMT states result from rigid-body movements (Fig. 3b and Supplementary Video 3), with no major structural rearrangements observed within the KpsE cage. In a nucleotide-free state, our cryo-EM analyses resolve two apo KpsMT conformations (Apo 1 and Apo 2) (Fig. 3a,b and Extended Data Fig. 6a,b). Apo 1 shows well-separated NBDs with minimal interactions mediated by TM5 between the KpsM subunits. In this conformation, the gap between TM1 and TM5 of opposing KpsM protomers creates a deep electropositive canyon that extends from the NBD into the transmembrane segment. The canyon ends near the periplasmic membrane surface at the interface between TM1 and a short periplasmic reentrant helix (PGH1) following TM5 of the opposing subunit (Fig. 3a,c,d and Extended Data Fig. 6b).

In the Apo 2 state, the NBDs are closer together and slightly shifted against each other (Fig. 3b, Extended Data Fig. 6a and Supplementary Video 3). A similar arrangement has been observed in the crystal structure of the isolated Wzt NBD dimer of the O-antigen ABC transporter[12]. Accordingly, the KpsM subunits move towards each other by about 12 Å as compared to the Apo 1 state, reducing the separation of the membrane-embedded subunits and narrowing the electropositive canyon (Extended Data Fig. 6b and Supplementary Video 3).

As expected, in the presence of ATP, the NBDs are closely associated with an Mg–ATP complex bound at each nucleotide-binding site (Fig. 3b, Extended Data Figs. 4 and 6 and Supplementary Video 3). The architecture of the KpsM dimer in the ATP-bound state resembles the Apo 1 conformation with an open electropositive canyon (Extended Data Fig. 6a,b). However, a notable difference exists at the cytosolic entrance to the canyon. The interface helix and the N-terminal segment of TM1 shift by about 5 Å towards TM5 of the opposing KpsM subunit, thereby narrowing the canyon (Fig. 3b).

Transitioning from the ATP to the ADP–AlF$_4^-$-bound KpsMT conformation includes an approximately 5-degree rigid-body anticlockwise

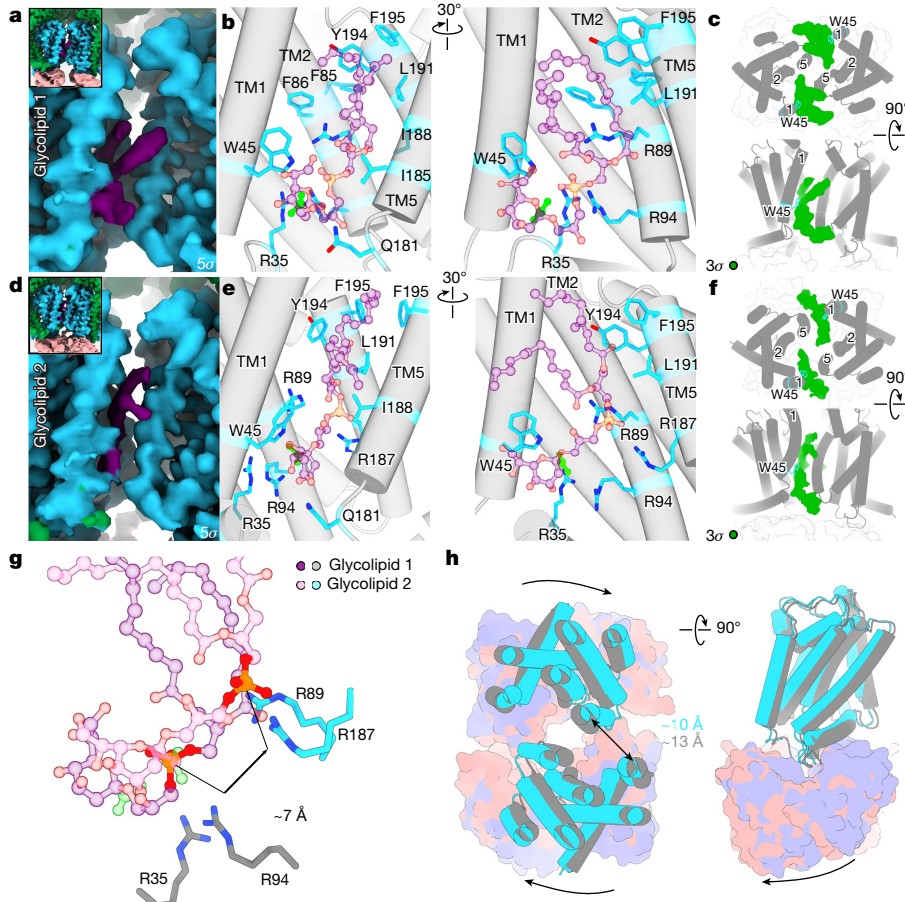

**Fig. 4 | Glycolipid loading into the KpsMT–KpsE complex. a**, Cryo-EM map of glycolipid 1, coloured magenta. The protein is coloured as in Fig. 2. **b**, Residues surrounding glycolipid 1, modelled as a Kdo–phosphatidylglycerol lipid molecule. The carboxyl group of Kdo is highlighted in lime for orientation. **c**, Low contour level of the map shown in **a** but viewed from the periplasm. Additional density (green) extends past the putative first Kdo moiety of the glycolipid. The KpsM dimer is shown as grey cylindrical helices with important transmembrane helices numbered. **d**–**f**, As in **a**–**c** but for glycolipid 2. **g**, Overlay of glycolipid 1 and 2 including two pairs of arginine residues that are likely to be involved in phosphate coordination. **h**, Conformational differences between glycolipid 1 (grey and purple) and glycolipid 2 (blue and salmon) bound states of KpsMT. Left, top-down view of KpsMT; right, side view of KpsMT. Cryo-EM densities in this figure are carved 2–3 Å from the models.

rotation (viewed from the periplasm) of the KpsM subunits around an axis running through the centre of TM2–TM4 of each subunit (Fig. 3b, Extended Data Fig. 6a and Supplementary Video 3). This movement narrows the distance between the opposing TM5 and widens the inter-protomer gap between TM1 and TM5.

## Glycolipid binding to the KpsM canyon

The in vivo CPS secretion experiments described above (Fig. 1e) show that the *St*ABC transporter recognizes and translocates a lipid-linked HA polymer formed by the *P. multocida* biosynthetic CPS components. We purified this CPS substrate from the engineered *E. coli* C43ΔCPS1 membrane fraction in a detergent-solubilized state using immobilized CBM70 (Methods). The obtained glycolipid is degraded by hyaluronidase and its electrophoretic mobility suggests a molecular mass of around 500 kDa, indicating a HA glycolipid of more than 1,000 disaccharide units (Extended Data Fig. 7a). Using the same purification approach, only background levels of HA were obtained from cells lacking KpsS or KpsC (Fig. 1a–e and Extended Data Fig. 7b). These enzymes produce the poly-Kdo linker connecting the phosphatidylglycerol lipid with the HA polymer, suggesting that the isolated glycolipid indeed represents the KpsMT substrate[7].

To determine how KpsMT recognizes the glycolipid, the purified KpsMT–KpsE complex was preincubated with the isolated glycolipid before cryo grid preparation (Methods). Cryo-EM analyses of this sample provided additional cryo-EM maps similar in overall architecture to the Apo 1 state, but with poorly defined KpsT subunits. KpsT was therefore placed after focused refinement and rigid-body docking (Methods and Extended Data Fig. 5).

In these structures, the above-described electropositive canyon at the interface between the KpsM subunits is open (Fig. 4). However, in contrast to all other structures, we identified two states of a lipid molecule bound inside the canyon (referred to as state-1 and state-2) (Fig. 4a–f and Extended Data Figs. 5 and 7). In state-2, the phosphate group of the lipid is shifted by about 7 Å towards the extracellular side, relative to state-1 (Fig. 4g).

Both lipid densities are connected to a globular density that readily accommodates a Kdo sugar (Fig. 4a–f and Extended Data Figs. 5 and 7). From this putative first Kdo residue, an additional weaker density extends past the interface of the KpsM subunits towards the cytosol and the KpsE carousel (Fig. 4c,f). Although the density could accommodate two to three additional Kdo units, the map quality is too weak for assignment. Therefore, we modelled the putative ligand as a phosphatidylglycerol-linked Kdo monosaccharide, with the caveat that the Kdo conformation and its interactions are not precisely resolved (Extended Data Fig. 5h–m). Of note, similar ligand densities are observed at the opposing KpsM dimer interface of the pseudo-twofold-symmetric transporter (Fig. 4c,f).

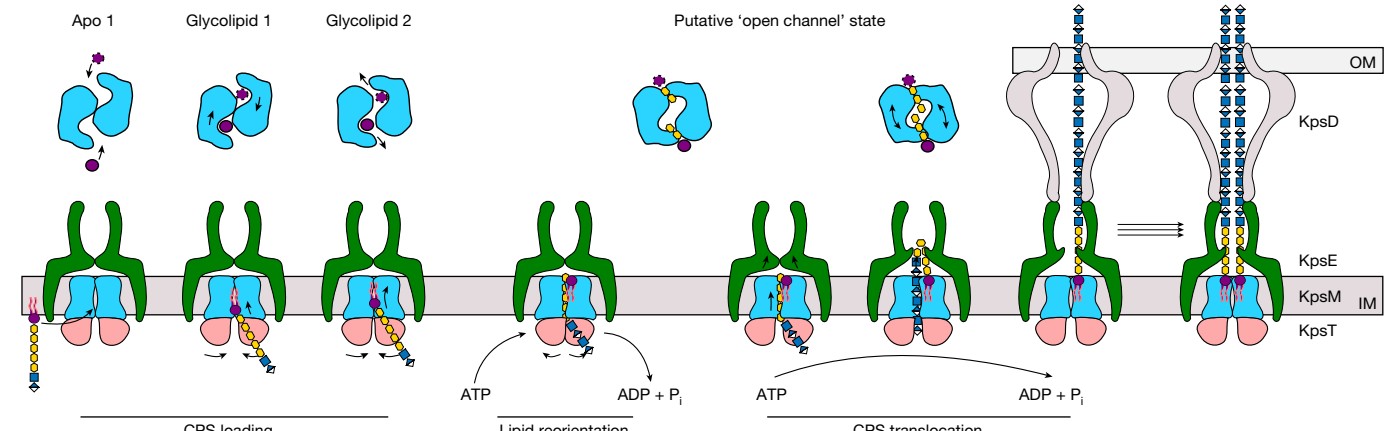

**Fig. 5 | Model of CPS translocation.** CPS secretion is likely to start with glycolipid loading into the KpsM canyon halfway across the membrane. Structural flexibility of KpsMT may facilitate different binding poses. The closure of the NBD after ATP binding may reorient the glycolipid in the inner membrane, thereby loading the CPS into the transporter proper. Sliding of the KpsM subunits against each other could create a transmembrane channel around the translocating CPS. Consecutive rounds of ATP hydrolysis translocate the CPS across the inner membrane and into the KpsE cage. CPS-induced rearrangements within the KpsE crown would then favour interactions with KpsD to form a trans-envelope conduit. The substrate's lipid moiety may exit laterally into the inner membrane. Multiple CPS polymers could be secreted by a single secretion system. IM, inner membrane; OM, outer membrane.

However, these densities are of lower quality and have not been interpreted.

In state-1, with the putative Kdo sugar closer to the cytosolic water–lipid interface, the sugar moiety is sandwiched between TM1 and TM5 of opposing KpsM protomers (Fig. 4b and Extended Data Fig. 7e). Although the exact position of the Kdo moiety is speculative, it is surrounded on one side by Trp45 of TM1 and, through the opposite face of the sugar ring, Gln181, Ile185 and Ile188 of TM helix 5 of the second KpsM protomer. The following phosphate group is near Arg94 of TM2 (Fig. 4b). All surrounding residues are conserved among CPS ABC transporters.

One lipid acyl chain snakes through the hydrophobic segment of the canyon, where it is surrounded by Phe85 and Phe86 of TM2, as well as by Leu191, Tyr194 and Phe195 of TM5 of the same protomer (Fig. 4b). The second acyl chain points away from the KpsM subunits and does not mediate extensive protein interactions.

In state-2, the lipid is shifted by about 7 Å towards the periplasm, such that the carboxyl group of the putative Kdo sugar is in proximity to Arg35 and Arg94 of the interface helix and TM2, respectively (Fig. 4d,e and Extended Data Fig. 7f). The lipid phosphate group contacts Arg187 of TM5 in the opposing KpsM subunit. Furthermore, at the C-terminal end of this helix, Gln181 is suitably positioned to interact with the Kdo moiety. Relative to the estimated cytosolic and periplasmic membrane boundaries, the glycolipid sits about halfway across the membrane, in agreement with it representing a translocation substrate (Fig. 4d,e).

Compared with state-1, the second lipid-binding pose further correlates with an approximately 6 Å rigid-body translation of the KpsM subunits against each other, thereby widening the space between the KpsM protomers and narrowing the entry to the polysaccharide canyon by around 3 Å (Fig. 4h). Extending this KpsM movement could create a channel-forming transporter conformation, as observed for the O-antigen WzmWzt transporter[12,32,33].

To test the functional relevance of the observed protein–glycolipid interactions, we altered canyon-lining residues in *St*KpsM (which are conserved in *Pm*KpsM) and monitored the ability of the mutated transporter to secrete the HA glycolipid in vivo (Extended Data Fig. 7i–k and Supplementary Fig. 24). None of the positively charged residues of the canyon (Arg35, Arg89 and Arg94) can be replaced with Lys or Ala without disrupting CPS export. Moreover, Trp45, juxtaposed with the putative Kdo moiety, cannot be replaced with Phe or Leu, underscoring its importance for CPS secretion. The generated mutants form stable KpsMT–KpsE complexes that can be copurified, suggesting that the subunits indeed fold and assemble properly (Extended Data Fig. 7i). The only exception was a mutation of Arg89 to Lys, which was unstable.

## Discussion

Several ABC transporters are known to secrete high-molecular-weight biopolymers, including polypeptides and polysaccharides. In an extended conformation, the polymers exceed the length of the transporters many times, probably requiring a stepwise translocation mechanism. CPSs, like unfolded proteins secreted by the bacterial type-1 secretion system (T1SS), are transported across the Gram-negative cell envelope in a single step. However, unlike the T1SS[34], a single CPS ABC transporter is fully encircled by an octamer of the periplasmic subunit KpsE.

The KpsE octamer provides an example of a class-3 PCP associated with an ABC transporter. Class 1 and 2 PCPs are associated with Wzy-dependent polysaccharide biosynthesis pathways and are likely to control the processivity of the polymerization reaction[35] (Extended Data Fig. 8). Although KpsE structurally resembles PCP-1 and PCP-2 in its crown and dome regions, class-2 PCPs contain a cytosolic tyrosine kinase domain that is implicated in phosphoregulation.

It is conceivable that the unresolved KpsE helices α6 and α7 at the tip of the oligomer interact with an octamer of KpsD. This CPS subunit is predicted by AlphaFold2 to assemble into a barrel, with its C-terminal domain probably traversing the outer membrane (Extended Data Figs. 1g and 8a). The ability to detect the KpsD C-terminal Strep tag on the cell surface supports this model. Thus, the translocating CPS is likely to be sequestered from the periplasm by the secretion system to avoid mislocalization. Because KpsE is dispensable for CPS translocation across the inner membrane in our engineered *E. coli* system, its probable biological function is to bridge the inner- and the outer-membrane-integrated components of the secretion system—similar to AcrA of tripartite efflux pumps[36,37]. The length of a predicted KpsMT–KpsE–KpsD complex is comparable with that of these envelope-spanning systems (Extended Data Fig. 8a).

The KpsMT–KpsE substrate is a lipid-linked polysaccharide. Other known ABC transporters involved in lipid transport across the bacterial inner membrane and/or the periplasm include the phospholipid-trafficking MlaFEDB and the LPS-extracting LptB2FGC complexes[38,39] (Extended Data Fig. 8a). None of these resemble the secretion-system-like KpsMT–KpsE or MacAB–TolC systems.

The KpsMT conformation in apo and nucleotide-bound states reveals neither a transmembrane channel nor intracellular or extracellular funnels that could accommodate the substrate polymer. Instead, the KpsM subunits form a narrow lipid-exposed canyon that might allow polysaccharide translocation. By analogy with other characterized CPS secretion systems, the translocation substrate contains a conserved region consisting of phosphatidylglycerol linked to a Kdo oligosaccharide, followed by a strain- and/or serotype-specific polysaccharide chain. On the basis of the observed cross-species complementation, it is likely that the transporters recognize the conserved lipid and/or polyanionic oligosaccharide region of the substrate to initiate secretion. The conserved electropositive canyon at the KpsM interface might serve this purpose.

Indeed, the lipid moiety of the isolated HA glycolipid binds to this location. The polysaccharide chain is likely to extend from the canyon towards the cytosolic segments of the KpsE cage, right above the interface of the KpsT subunits (Fig. 5). As observed in vitro, CPS secretion could start with spontaneous substrate binding to the KpsM canyon, facilitated by electrostatic interactions and shape complementarity (Fig. 5 and Extended Data Fig. 9). The two resolved glycolipid-bound states suggest different ligand-binding sites and/or ATP-independent migration of the substrate within the canyon. Because KpsM is structurally similar to Wzm of the O-antigen ABC transporter, it is possible that KpsMT adopts a similar channel-forming conformation to that of WzmWzt during CPS export[12,33] (Fig. 5 and Extended Data Fig. 9).

As the substrate migrates further through the transporter, the lipid headgroup may reorient in the membrane and exit the transporter laterally into the periplasmic leaflet of the inner membrane. A potential lateral gate exists between TM1 and the PGH1 reentrant helix of the opposing KpsM subunit. In agreement with the pseudo-twofold symmetry of KpsMT and the observed ligand densities on both KpsM dimer interfaces, the simultaneous translocation of two substrate molecules is conceivable.

Our data suggest that CPS translocation starts with the lipid moiety and not the terminal CPS glycosyl unit. Whether the lipid anchor is extracted from the inner membrane and integrated into the outer membrane, or whether it remains in the inner membrane while the CPS spans the cell envelope is unknown. Our structural data do not suggest a mechanism for lipid extraction after reorientation in the inner membrane. Instead, the lipid might dissociate from the transporter after reorientation, with the following CPS being pushed into the KpsE–KpsD trans-envelope tunnel (Fig. 5). The volume of the KpsE octamer and the space surrounding the ABC transporter inside the KpsE cage would allow several CPS molecules to accumulate. The disordered KpsE L2 loop extending into the KpsE tunnel would have to reposition to accommodate the translocating CPS.

The CPS translocation substrate isolated from the engineered *E. coli* suggests an HA length exceeding 1,000 disaccharide repeat units, in agreement with CPS sizes isolated from uropathogenic *E. coli*[40]. With an estimated distance of about 10 Å per repeat unit, this correlates with a polymer length of approximately one micrometre—about half the length of *E. coli*. The patched distribution of KpsD in the outer membrane and uniform CPS surface labelling revealed by dual-colour MINFLUX nanoscopy suggest that CPS coverage can be achieved from limited CPS secretion sites, indicating spreading of the polysaccharides on the cell surface. Future detailed insights into a complete CPS secretion system during translocation and its cellular dynamics will be necessary to derive a complete model of capsule biogenesis.

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

## Methods

### DNA manipulation

**Plasmid design for *Pm*CPS genes.** All *Pm*CPS genes were codon optimized for *E. coli* expression, synthesized by Bio Basic and cloned into pETDuet and pACYCDuet plasmid backbones, giving rise to expression plasmids 5 and 6, pETDuet Hex Secretion System and pACYCDuet Biosynthesis components, respectively (see 'Plasmids' in the Supplementary Information). Plasmid 6 encodes KpsC (with an N-terminal Strep-Tag II), KpsS (N-terminally 8×His-tagged), HyaE (N-terminally Myc-tagged), HyaB (N-terminally Flag-tagged) and HyaD (N-terminally 8×His tagged), each under a separate T7 promoter and lac operator. Plasmid 5 encodes KpsD (with a C-terminal Strep-Tag II), KpsE (N-terminal S-tag), KpsM (untagged) and KpsT (C-terminally 8×His tagged), each under a separate T7 promoter and lac operator. In both plasmids, each CPS gene is flanked by a unique restriction site. Single-gene deletions for in vivo encapsulation assays were achieved by restriction enzyme (NEB) digest and religation using T4 ligase (NEB). The restriction enzymes used in single-gene deletions from plasmid 6 were NcoI (KpsC), BamHI (KpsS), SacI (HyaE), EcoRI (HyaB) and XhoI (HyaD); and for plasmid 5 they were NdeI (KpsD) and BamHI (KpsE).

**Two-plasmid genome-editing system.** Plasmid design was based on a previous study[42], in which efficient *E. coli* genome editing was achieved using a two-plasmid approach: plasmid 7 (pCasJZ) and plasmid 8 (pUC57_region1_tet). Plasmid 7 encodes *Sp*Cas9 under an arabinose promoter, the red λ phage proteins Gam, Beta and Exo under a T7 promoter, the SacB cassette and the chloramphenicol resistance cassette. Plasmid 8 encodes two 500-bp-long homology regions (1 and 2), flanked by protospacer adjacent motifs (PAMs) (1 and 2) and target sequences (1 and 2), two guiding RNA sequences with target sequences (1 and 2) and the tetracycline resistance locus. PAMs 1 and 2 and the target sequences 1 and 2 were chosen to be upstream and downstream, respectively, of CPS region 1 in the *E. coli* C43 genome. Homology regions 1 and 2 are 500 bp sequences upstream and downstream of target sequences 1 and 2.

Plasmid 7 was generated using the Gam, Beta, Exo, λ tL3 terminator, SacB, SacB promoter, araBAD promoter and *Sp*Cas9 elements from plasmid pCasPA (Addgene 113347), which were cloned by Gibson Assembly (NEB) into MCS-1 of the pACYCDuet plasmid. Next, *araBAD* and *araC* were cloned in place of the deleted MCS-II of that plasmid. The homology regions, guiding RNAs, PAMs and target sequences were synthesized by Gene Universal into the pUC57 vector. Then, using Gibson assembly, the ampicillin resistance cassette was replaced with the tetracycline resistance cassette from pBR322 plasmid, giving rise to plasmid 8.

**Genome editing.** The procedure has been described in detail elsewhere[42]. In brief, plasmids 7 and 8 were co-transformed into *E. coli* C43 (DE3). The working concentrations of isopropyl-β-D-thiogalactopyranoside (IPTG), arabinose, chloramphenicol, tetracycline, glucose and sucrose were 1 mM, 20 mM, 25 mg l⁻¹, 10 mg l⁻¹, 1% and 2%, respectively. Cells were plated on LB agar plates, supplemented with chloramphenicol, tetracycline and glucose. Single colonies were used to inoculate 2 ml of LB medium supplemented with chloramphenicol, tetracycline and glucose. After 2 h of growth at 37 °C, IPTG was added, and the expression of red λ phage proteins was induced. After an another 1 h, arabinose was added and the expression of *Sp*Cas9 and transcription of sgRNAs was induced. After 3 h, cells were plated on LB agar plates containing chloramphenicol, tetracycline and arabinose and incubated overnight at 37 °C. Positive colonies were verified by colony PCR using primers flanking the 500-bp-long homology regions and sequencing (Supplementary Figs. 2 and 3). Then, the positive clone was grown in LB containing sucrose and tetracycline to remove plasmid 7. No chloramphenicol-resistant colonies were detected.

Genome-edited C43 cells lacking the CPS1 region, termed C43ΔCPS1, were made electrocompetent and used in the in vivo encapsulation assay.

**Plasmid design for CBM70 and SNAP–CBM70.** The codon-optimized gene encoding *Streptococcus pneumoniae* carbohydrate-binding module 70 (CBM70)[20], was synthesized by Bio Basic and cloned into the pET30 vector with a C-terminal 10×His tag, generating plasmid 9. Then, using Gibson Assembly, the SNAP tag from the pSNAP-tag vector (Addgene 101135) was N-terminally fused to CBM70 with a GSSMGS linker, creating plasmid 10.

**Plasmid design for *St*KpsMT-E-D.** The codon-optimized KpsD, KpsE, KpsM and KpsT genes from *S. thermodepolymerans* were synthesized and cloned into expression vectors by Gene Universal, generating plasmids 1a, 3a and 4. Plasmid 1a contains KpsM in MCS-I and KpsT (C-terminal Flag) in MCS-II in a pETDuet backbone. Plasmid 3a contains KpsE (C-terminal 10×His tagged) in MCS-I in a pCDFDuet backbone. Plasmid 4 contains KpsD in a pET30 backbone. To increase the expression yields of KpsM and KpsE, we introduced three amino acids as the third to the fifth residues of the polypeptide chains ($K_3$-$I_4$-$H_5$) (using polymerase incomplete primer extension (PIPE) cloning) that were shown to increase translation initiation[43]. KpsE was further modified by introducing two cysteines at positions 77 and 138, giving rise to plasmids 1b and 3b. Next, using Gibson assembly, KpsD was inserted into MCS-II of plasmid 3b, creating plasmid 3c, used for in vivo encapsulation assays.

**Mutagenesis.** Mutagenesis was performed by PIPE cloning with overlapping primers on plasmids 1b and 3b, using Phusion HF DNA polymerase (NEB), resulting in plasmids 17–24.

### Protein and CPS expression

**Bacterial growth.** All bacterial cultures described in this work were grown at 37 °C and with shaking at 220 rpm, unless noted otherwise. Working concentrations: ampicillin 100 mg l⁻¹, kanamycin 50 mg l⁻¹, streptomycin 50 mg l⁻¹ and chloramphenicol 25 mg l⁻¹. Appropriate plasmids were transformed into C43 cells (for protein purification) or C43ΔCPS1 cells (for in vivo encapsulation, MINFLUX and CPS purification) for overnight growth in the presence of suitable antibiotics. All collected cell pellets in this study were flash-frozen in liquid nitrogen and stored at −80 °C for further use.

For western blotting: For expression testing, an overnight starter culture of cells expressing all *Pm*CPS components was used to inoculate 1 l of LB medium supplemented with ampicillin and chloramphenicol. At an optical density at 600 nm (OD₆₀₀) of 0.6, protein expression was induced with 100 mg l⁻¹ of IPTG. Growth was continued for another 3–4 h, after which cells were collected (4,500 rpm for 20 min) and flash-frozen in liquid nitrogen. This cell pellet was used to prepare inverted membrane vesicles, as described previously[44]. The inverted membrane vesicles were then run on a 12.5% polyacrylamide gel and analysed using the western blot technique detecting the engineered affinity tags, as described previously[45].

For in vivo encapsulation assays and spheroplasting: 20 ml LB culture was inoculated with a single stab of the appropriate transformants and grown in the presence of appropriate antibiotics and 100 mg l⁻¹ IPTG. Growth was carried out for 6–8 h, after which cells were collected. Spheroplasts were prepared as described previously[46]. After removal of the outer membrane, spheroplasts were resuspended in PBS supplemented with 200 mM sucrose, and used for in vivo encapsulation assays.

For purification of CPS: 8×1 l of 2× LB supplemented with appropriate antibiotics were inoculated from an overnight starter culture. At an OD₆₀₀ of 0.6, the medium was cooled to 20 °C, and then cells were induced using 100 mg l⁻¹ of IPTG, grown overnight and collected.

For purification of the CBM70, SNAP–CBM70 and KpsMT–KpsE proteins: 6× 1 l of 2× LB supplemented with appropriate antibiotics were inoculated from an overnight starter culture. At an $OD_{600}$ of 0.6, protein expression was induced using 200 mg $l^{-1}$ of IPTG, and cells were grown for another 3–4 h at 37 °C, after which they were collected.

### In vivo encapsulation assay

A total quantity of 200 µl of cells at $OD_{600}$ of 4 was washed with ice-cold PBS three times and then incubated on ice with 10 µl of 2 mg $ml^{-1}$ of [Alexa647]SNAP–CBM70 or [Flux680]CBM70 (see below) for a total of 2 h. For CPS digestion, samples were treated with 1 mg $ml^{-1}$ of bovine testicular hyaluronidase (MP Biomedicals) for 2 h on ice. For two-colour MINFLUX nanoscopy, 12.5 µl of [Alexa680]Streptavidin (Thermo Fisher Scientific) was added after 1 h of incubation, followed by incubation on ice for 30 min. Then 2 µl of 100× Cellbrite Fix 488 (Biotum) was added for another 30 min, after which cells were washed three times in 1,000 µl of ice-cold PBS, fixed with 4% PFA (Electron Microscopy Sciences) in a total of 1,000 µl PBS for 20 min, blocked with 50 mM $NH_4Cl$ in PBS for 30 min, resuspended in 200 µl ice-cold PBS and imaged.

**Confocal microscopy.** Imaging was performed on a Zeiss LSM880 confocal microscope with an Airyscan detector, at 40× using a water-immersion objective. Membrane and CPS channels were recorded sequentially using 488-nm and 633-nm excitation lasers, respectively, and suitable filter sets. The pixel size was set to 52 nm. Images were processed in ImageJ–Fiji[47].

### Metabolic labelling of LPS with AZ6470 DBCO

Metabolic labelling of the LPS was achieved following the protocol described previously[23]. In brief, cells expressing all necessary components for CPS biosynthesis were grown for 8 h at 37 °C in the presence of 1 mM Kdo azide (Click Chemistry Tools) and 100 mg $l^{-1}$ IPTG. After that, cells were washed three times and resuspended in M9 medium. Then AZDye647 DBCO (Click Chemistry Tools) was added to a final concentration of 0.5 µM for the copper-free click chemistry reaction. Cells were incubated in the dark for another 1 h at 37 °C with shaking. Finally, the cells were washed three times with PBS, aliquoted, flash-frozen in liquid nitrogen and stored at −80 °C.

### Sample mounting, imaging buffer and nanoscopy for MINFLUX.

Fixed cells were applied on glass slides precoated with poly-L-lysine. Samples were mounted in the imaging buffer as described[48]. In brief, gold nanoparticles (Nanopartz, A11-200-CIT-DIH-1-10) were used as fiducials. GLOX buffer supplemented with 10–14 mM MEA (Cysteamine) was used as imaging buffer. Samples were sealed with EliteDouble22 (Zhermack).

MINFLUX nanoscopy and corresponding confocal microscopy were performed using an in-house MINFLUX set-up[48]. In two-colour MINFLUX, a single event originates from one of two different red fluorophores (Alexa 647/AZdye647 or Alexa 680/Flux680) and is split into Cy5-near and Cy5-far detectors. The number of photons from this single event reaching both detectors is represented as a detector channel ratio (DCR) and is characteristic for each emitter. Experimental DCR values were separately acquired for each fluorophore, and then used to assign the colour of fluorophores to localizations in two-colour MINFLUX experiments. Pixel size in rendered images was based on localization precisions of raw burst.

### Protein purification

When possible, all purification steps were performed at 4 °C.

**CBM70 and SNAP–CBM70 purification.** Cell pellets were thawed and resuspended in 10 % glycerol, 100 mM NaCl, and 20 mM Tris pH 7.5, then incubated for 1 h with 1 mg $ml^{-1}$ lysozyme. PMSF (1 mM) was added and cell suspensions were lysed by three passes through a

microfluidizer. Intact cells were removed by low-speed centrifugation for 25 min at 12,500 rpm, in a JA-20 rotor (Beckman). The supernatant was centrifuged for 1 h at 200,000g in a Ti45 rotor (Beckman) and the insoluble material was discarded. The supernatant was spiked with 20 mM imidazole and incubated with Ni-NTA resin for 1 h with agitation. The resin was washed with (1) 1 M NaCl, PBS (pH 7.4) and 40 mM imidazole and (2) PBS and 60 mM imidazole. Protein was eluted after a 30-min incubation in PBS containing 320 mM imidazole, and was concentrated using a 10-kDa filter (Amicon) to 1 ml. Then, in the case of CBM70, the sample was run over an S200 16/60 gel filtration column equilibrated in PBS and the peak fractions were collected and used for αHA column preparation, or aliquoted and flash-frozen for storage; or, in the case of SNAP–CBM70, dialysed against PBS overnight, aliquoted, flash-frozen in liquid nitrogen and stored at −80 °C.

**Purification of KpsMT–KpsE.** Cell pellets were processed as described above. After low-speed centrifugation, membranes were isolated from the lysate by centrifugation for 2 h at 200,000g in a Ti45 rotor, then collected, flash-frozen in liquid nitrogen and stored at −80 °C.

Membranes were thawed and resuspended in 300 mM NaCl, 20 mM Tris pH 7.5, 10 % glycerol, 40 mM imidazole, 1% n-dodecyl-β-maltoside (DDM) and 0.1% cholesterol hemisuccinate and incubated for 1 h with agitation. Aggregated material was removed by centrifugation at 200,000g for 30 min, and the supernatant was incubated with Ni-NTA resin for 1 h. The resin was washed with (1) 1.5 M NaCl, 20 mM Tris pH 7.5, 10 % glycerol, 40 mM imidazole and 0.1% LMNG and (2) 300 mM NaCl, 20 mM Tris pH 7.5, 10% glycerol, 80 mM imidazole and 0.1% LMNG. Protein was eluted after a 30-min incubation in 300 mM NaCl, 20 mM Tris pH 7.5, 5% glycerol, 400 mM imidazole and 0.05% LMNG, and was concentrated to 500 µl using a 100-kDa filter (Amicon), followed by overnight incubation on ice. The next day, the sample was run over a S6-increase 10/300 gel filtration column equilibrated in 100 mM NaCl, 50 mM Tris pH 7.5 and 0.025% LMNG. This buffer was supplemented with 5 mM $MgCl_2$ for KpsMT(E151Q)-KpsE preparations. The peak fractions were collected and concentrated to 2–3 mg $ml^{-1}$ using a 100-kDa filter, and were used for grid preparation or in vitro ATPase activity assays.

KpsMT-KpsE in complex with ADP–AlF$_4^-$ was purified similarly. The concentrated Ni-NTA elution sample was dialysed overnight against buffer containing 100 mM NaCl, 50 mM Tris pH 7.5, 0.05% LMNG, 5% glycerol, 10 mM NaF, 2 mM $AlCl_3$, 5 mM ADP and 5 mM $MgCl_2$. The same buffer containing 0.025% LMNG and lacking glycerol was used to equilibrate the S6-increase gel filtration column. Peak fractions were collected on the basis of elution times, concentrated to around 2–3 mg $ml^{-1}$ and used for cryo grid preparation.

### Preparation of Alexa647-tagged SNAP–CBM70 and Flux680-tagged CBM70

All steps were performed in the dark at 4 °C. SNAP–CBM70 aliquots were thawed and mixed with dimethyl sulfoxide (DMSO)-solubilized Alexa647 (NEB) in a 1:1 molar ratio, in the presence of 1 mM DTT. The sample was incubated with agitation for 6–8 h and run over an S200 10/30 gel filtration column equilibrated in PBS to separate [Alexa647]SNAP–CBM70 from the free dye. Peak fractions with string absorbances at 280 and 671 nm were aliquoted at 2 mg $ml^{-1}$, flash-frozen in liquid nitrogen and stored at −80 °C.

[Flux680]CBM70 (lacking the SNAP domain) was prepared in a similar manner. CBM70 aliquots were thawed and mixed with DMSO-solubilized Flux680-maleimide (Abberior) in a 1:3 molar ratio, in the presence of 1 mM DTT. The sample was incubated with agitation for 16 h and run over an S200 10/30 gel filtration column equilibrated in PBS to separate [Flux680]CBM70 from the free dye. Peak fractions with string absorbances at 280 and 695 nm were aliquoted at 2 mg $ml^{-1}$, flash-frozen in liquid nitrogen and stored at −80 °C.

## ATPase activity assays

The ATPase activity of KpsMT-KpsE was quantified using an enzyme-coupled assay as previously described[32]. The peak fraction of the complex eluting from a S6-increase column was concentrated to 0.5–1 mg ml$^{-1}$ and used for activity assays. ATPase activity was initiated by adding ATP, and the depletion of NADH was monitored at 340 nm for 1 h at 27 °C in a SpectraMax M5 plate reader. The rate of NADH depletion was converted to nmol ATP hydrolysed using an ADP standardized plot. The data were processed in Microsoft Excel and GraphPad Prism. All experiments were performed at least in triplicate and error bars represent deviations from the means.

## Anti-hyaluronan affinity column preparation

Purified CBM70 was coupled to NHS-activated Sepharose 4 Fast Flow beads (Cytiva) following the manufacturer's protocol. In brief, the resin was washed with (1) MQ water, (2) 1 mM HCl and (3) PBS. Then, 20 ml of 5 mg ml$^{-1}$ CBM70 in PBS was mixed with 25 ml of the washed resin, and left agitating for two days at 4 °C. After that, the liquid was drained from the beads, and the resin was washed with PBS, followed by blocking buffer (PBS containing 200 mM ethanolamine) for 24 h with agitation at 4 °C. The beads were washed and stored in 20% ethanol at 4 °C.

## CPS purification

Cell pellets and membranes were prepared as described above. Membranes were resuspended in PBS containing 1% LMNG and incubated for 1 h at room temperature with agitation. Aggregated material was removed by centrifugation at 200,000$g$ for 30 min, and the supernatant was incubated with anti-HA resin (see above) for 1 h. Next, the beads were washed three times with PBS containing 0.1% LMNG. The CPS was eluted from the column after a 30-min incubation with 2 M NaCl, 100 mM sodium citrate pH 3.0 and 0.01% LMNG, concentrated to 250 µl using a 3-kDa filter (Amicon) and dialysed overnight against PBS in a 3.5-kDa dialysis membrane. The next day, the sample was run on a 1.5% agarose gel (Ultra-pure agarose, Invitrogen) or a 4–20% gradient polyacrylamide gel (Bio-Rad), and stained with Stains-All dye (Sigma) as described[49]. The obtained CPS sample was also used for cryo-EM analyses.

## HA ELISA assay

The total amount of cell-surface-exposed HA was quantified using an ELISA-based kit (Echelon Biosciences) following the manufacturer's protocol. Cell densities were adjusted on the basis of OD$_{600}$. HA was quantified on chemiluminescence using a Promega GloMax plate reader. The data were processed in Excel and GraphPad Prism. All experiments were performed at least in triplicate.

## Grid preparation and data collection

To obtain the ATP- and glycolipid-bound states, KpsMT(E151Q)-KpsE and wild-type KpsMT-KpsE were supplemented with 2 mM ATP or 30 µl of lipid-linked HA, respectively, before grid preparation. Quantifoil holey carbon grids (Cu 1.2/1.3, 300 mesh) were glow-discharged in the presence of two drops (about 200 µl) of amylamine. Four microlitres of sample was applied, blotted for 4–10 s with a blot force of 4–7 at 4 °C and 100% humidity, then plunge-frozen in liquid ethane using a Vitrobot Mark IV (FEI).

Cryo-EM data were collected at the University of Virginia Molecular Electron Microscopy Core (MEMC) on a Titan Krios (FEI) 300-kV electron microscope using a Gatan Imaging Filter (GIF) and a K3 direct electron detection camera. Movies were collected in EPU (Thermo Fisher Scientific) at a magnification of 81,000× with an energy filter width of 10 eV, using counting mode with a total dose of 51 e$^-$ per Å over 40 frames, and with a target defocus of −1.0 to −2.0 µm.

## Cryo-EM data processing

All datasets were processed in cryoSPARC (versions 3.3 and 4.0)[50]. Raw movies were subjected to patch motion correction and patch contrast transfer function (CTF) estimation. For all four datasets, particles were automatically selected by blob picker to generate initial templates, followed by template picker. After several rounds of 2D classification, selected particles were used for ab initio reconstructions in $C1$, followed by heterogenous refinement.

For dataset 1 yielding the Apo 1 state, $C2$ symmetry was applied in both non-uniform and local refinements. To improve the KpsT density, focused three-dimensional (3D) classification followed by non-uniform and local refinement was applied. Using the Phenix Combine Focused Maps job[51], a composite map 1 was created from maps A and B, on the basis of the model and half maps from the focused refinements.

For dataset 2 yielding the ATP-bound state, $C2$ symmetry was applied during non-uniform refinement, followed by local refinement in $C1$. To improve the density of the crown region of KpsE, focused 3D classification followed by non-uniform and local refinement was applied. Using Phenix Combine Focused Maps job, a composite map 2 was created from maps A and B, on the basis of the model and half maps from the focused jobs.

For dataset 3 yielding the ADP–AlF$_4$$^-$-bound state, $C2$ symmetry was applied in non-uniform refinement.

For dataset 4 yielding the glycolipid 1 and 2 and Apo 2 states, $C1$ symmetry was applied in both non-uniform and local refinements. Next, the particles were 3D classified into seven classes using seven identical Apo 1 volumes as input. Three of the resulting classes (class 0, 1 and 4) revealed distinct states and were subjected to non-uniform refinement. Classes 0 and 1 had noticeable extra density in the polysaccharide canyon. To improve the KpsT density in classes 0 and 1, a separate 3D classification focused on KpsT was performed, resulting in improved KpsT density for the glycolipid 1 and 2 states (respective map A for both classes), enabling rigid-body docking of an AlphaFold2-predicted KpsT model. Particles from classes 0 and 1 were also subjected to 3D classification focused on the KpsM subunits, which resulted in improved glycolipid density maps (respective map B for both classes). Then, the maps focused on KpsT and KpsM (respective maps A and B for both classes) were combined, resulting in composite maps 4 and 5 for glycolipids 2 and 1, respectively.

For this dataset, class 4 from the original 3D classification job revealed a novel arrangement of the KpsM transmembrane helices. As for classes 0 and 1, the KpsT map quality was improved by 3D focused classification, followed by non-uniform refinement. The improved map allowed rigid-body docking of a KpsT model. The focused maps A and B were combined using Phenix Combine Focused Maps job, resulting in map 6. Maps were sharpened on the basis of models using either the autosharpen (maps 1, 2, 3 and 6) or the local anisotropic sharpening (maps 4 and 5) jobs in Phenix:refine. Half maps were used to generate global-resolution estimates using EMBL's Fourier shell correlation (FSC) server, and local-resolution estimates using cryoSPARC's local resolution estimation job.

In all datasets, we observed a minor particle population with an incomplete KpsE cage. In these cases, the disordered eighth KpsE subunit is proximal to the interface of the KpsM subunits.

## Model building and refinement

To generate the initial model of the Apo 1 state of KpsMT-KpE, the Alphafold2 models[25] of the individual subunits were rigid-body-docked into the EM map using Chimera[52], and the model was iteratively real-space refined in Coot and Phenix:refine[51,53]. The obtained structure was used to build all other states. Chain completeness for all states is reported in 'Chain completeness' in the Supplementary Information. For Apo 2 and the glycolipid 1 and 2 states, two sets of real-space refine jobs were run, with and without the KpsT subunits rigid-body-docked into the model (Extended Data Table 1). For glycolipid-bound states 1 and 2, the putative substrate model was drawn using Coot Ligand Builder, on the basis of 1,2-dipalmitoyl-phospatidylglycerol and four Kdo sugars linked by β-2-7 and β-2-4 glycosidic linkages. The SMILES output from this job

was used in Phenix eLBOW to generate coordinate and constraint files. To preserve the correct linkage geometry connecting the first two Kdo units during refinement, the generated poly-Kdo-phosphatidylglycerol lipid was trimmed to two Kdo sugar units, docked into the substrate density in Coot and real-space-refined in Phenix.refine. After this refinement, the second Kdo sugar was removed from the substrate and the whole model was refined again in Phenix.refine. After the glycolipid 1 and 2 models were refined, the corresponding maps were locally anisosharpened in Phenix[54], resulting in maps 4 and 5. All structural figures were prepared using ChimeraX[55] and Inkscape (https://inkscape.org).

**AlphaFold2 predictions.** The full-length KpsE octamer was predicted at servers of the University of Virginia Molecular Electron Microscopy Core. *Pm*KpsD and *St*KpsD were predicted on AlphaFold2 Collab servers using truncated protein sequences (no outer membrane signal sequence, N-terminal truncation), to limit the octameric protein sequence to fewer than 3,300 amino acids. Omitted regions were then backfitted on the predicted octameric backbone from a monomer model.

### Reporting summary

Further information on research design is available in the Nature Portfolio Reporting Summary linked to this article.

### Data availability

The atomic coordinates of KpsMT–KpsE in the Apo 1, ATP-bound, ADP–AlF$_4^-$-bound, Apo 2, glycolipid 1 and glycolipid 2 states have been deposited at the Protein Data Bank (PDB) under accession codes 8TSW, 8TSH, 8TSI, 8TSL, 8TUN and 8TT3, respectively. Cryo-EM densities for the consensus maps of KpsMT–KpsE in the Apo 1, ATP-bound, ADP–AlF$_4^-$-bound, Apo 2, glycolipid 1 and glycolipid 2 states have been deposited at the Electron Microscopy Data Bank (EMDB) under accession codes EMD-41601, EMD-41592, EMD-41593, EMD-41595, EMD-41626 and EMD-41602, respectively.

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

**Acknowledgements** We thank P. Azadi and A. Muszynski for their efforts with glycolipid characterization; M. Purdy, K. Dryden and R. Ho for help with cryo-EM data collection; S. Schnorrenberg and T. Zimmermann for MINFLUX training; C. Whitfield, A. Gahlmann, V. Kiessling and J. Matthias for discussions; Z. Stephens, I. Górniak and F. Maloney for comments on the manuscript; and L. Tamm for use of the fluorescence microscope. J.K. is a recipient of the University of Virginia's Wagner graduate student fellowship and J.Z. acknowledges funding from the University of Virginia's Pinn Scholarship program to initiate the project. The first phase of the project was funded by NIH grant R21AI1642, proceeded by R35GM144130 awarded to J.Z. J.Z. is an investigator of the Howard Hughes Medical Institute (HHMI). This article is subject to HHMI's Open Access to Publications policy. HHMI laboratory heads have previously granted a nonexclusive CC BY 4.0 licence to the public and a sublicensable licence to HHMI in their research articles. Pursuant to those licences, the author-accepted manuscript of this article can be made freely available under a CC BY 4.0 licence immediately upon publication.

**Author contributions** J.K. performed all experiments. J.K. and J.Z. analysed the data and wrote the manuscript.

**Competing interests** The authors declare no competing interests.

**Additional information**
**Correspondence and requests for materials** should be addressed to Jochen Zimmer.

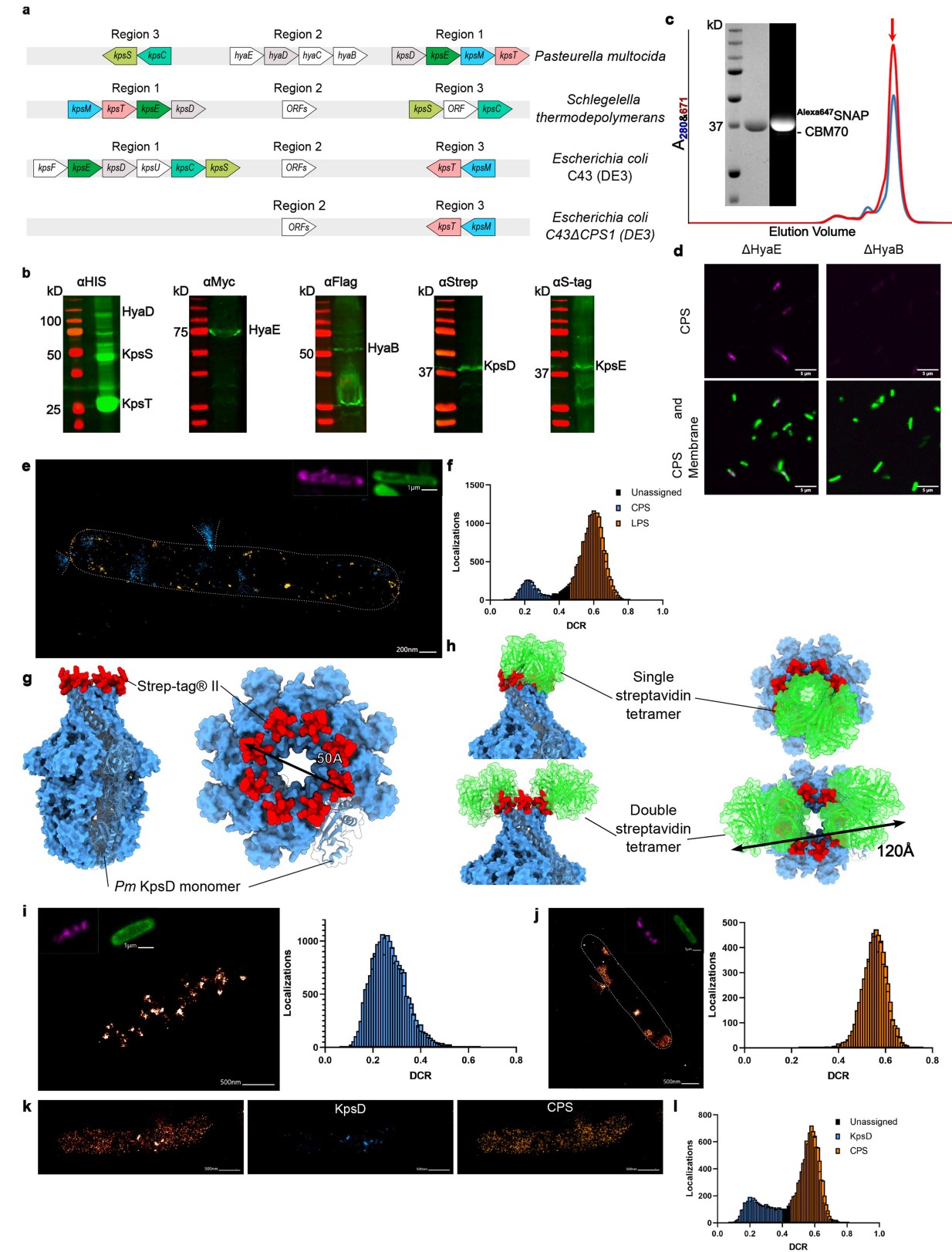

**Extended Data Fig. 1** | See next page for caption.

**Extended Data Fig. 1 | Genetic organization and recombinant expression of CPS components. a**, Genomic organization of CPS loci in different organisms. *E. coli* C43ΔCPS1 was created in this study using the CRISPR–Cas9/recombineering method. ORFs: open reading frames without assigned function. **b**, Recombinant expressions of *Pm* CPS components in *E. coli*. Shown are western blots of inverted membrane vesicles containing the individually tagged, membrane-bound CPS components. **c**, Alexa647 loading of the in-house purified SNAP–CBM70 probe. Size-exclusion chromatography profile for protein absorbance (280 nm - blue) and fluorophore absorbance (671 nm - red). The inset shows a Coomassie stained gel (left) and in gel fluorescence at 671 nm emission (right). **d**, Confocal images of *E. coli* C43ΔCPS1 cells expressing the *Pm* CPS components in the absence of HyaE or HyaB. CPS was stained as in Fig. 1 using the [Alexa647]SNAP–CBM70 probe, membranes were stained with Cellbrite Fix 488. **e**, Representative dual-colour MINFLUX nanoscopy of an encapsulated cell with LPS metabolic labelling. CPS (blue) is labelled with [Flux680]CBM70, LPS (orange) is labelled with AZ647 DBCO. Volcano-like CPS scattering and the cell outline are indicated with a dashed line. **f**, Histogram of DCR (detector channel ratio) values used for colour assignment in Extended Data Fig. 1e. Unassigned: localizations of overlapping DCR values that cannot be clearly assigned to either AZ647 or Flux680. **g**, Surface/cartoon representation of the AlphaFold2-predicted octameric *Pm* KpsD structure (blue) with the likely localization of the engineered C-terminal Strep-tag peptide (red). **h**, Possible configurations of tetramers of [Alexa680]streptavidin (PDB: 6j6j) bound to a KpsD octamer. A complex of a KpsD octamer bound to two [Alexa680]streptavidin tetramers would create a fluorophore cloud of about 12 nm diameter. **i**, MINFLUX nanoscopy of KpsD in an encapsulated cell via [Alexa680]streptavidin labelling (left panel). Most localizations were detected via the Cy5-far detector (DCR < 0.5) (right panel). **j**, MINFLUX nanoscopy of CPS of an encapsulated cell using [Alexa647]SNAP–CBM70 (left panel). Volcano-like CPS scattering and the cell outline are indicated with dashed lines. Most localizations were detected via the Cy5-near detector (DCR > 0.5) (right panel). For panels i and j, the experiments were repeated at least 3 times with similar results. **k**, Left: Dual-colour MINFLUX nanoscopy localizations of the encapsulated cell presented in Fig. 1f with [Alexa647]SNAP–CBM70 labelled CPS and [Alexa680]streptavidin labelled KpsD without fluorophore assignment, middle: only Alexa680 localizations coloured, right: only Alexa647 localizations coloured according to the colour assignment presented in **j**. **l**, Histogram of DCR values used for the fluorophore assignment in Fig. 1f. Unassigned: localizations of overlapping DCR values that cannot be clearly assigned to either Alexa647 or Alexa680.

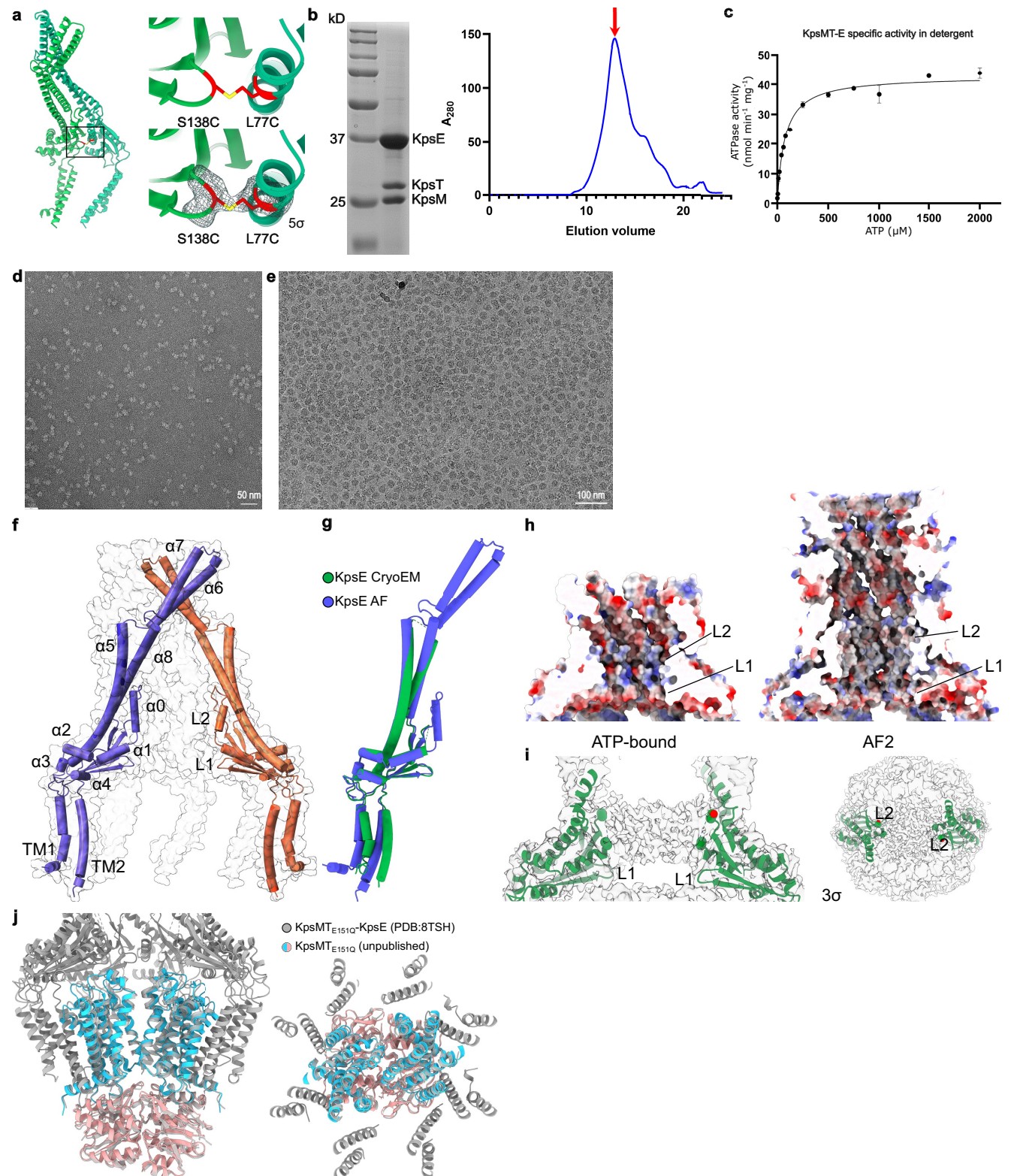

**Extended Data Fig. 2 | KpsE stabilization, purification, cryo grid preparation and AlphaFold2 prediction. a**, Cartoon representation of an AlphaFold2 KpsE dimer model used to engineer an intermolecular disulfide bridge. Electron density of the engineered disulfide bond observed in the cryo-EM map of the ATP-bound state. **b**, Size-exclusion chromatography profile of the KpsMT–KpsE complex (right) and peak fraction run on a reducing SDS-PAGE Coomassie stained gel (left). **c**, ATPase activity of the purified KpsMT–KpsE complex determined by monitoring the release of ADP via an enzyme-coupled assay. **d,e**, Negative stain and cryo-EM micrographs of the KpsMT–KpsE complex.

**f**, An AlphaFold2-predicted full-length *St* KpsE model indicating the localizations of three missing helices: α0 within the L2 loop and α6 and α7 of the crown. **g**, Overlay of cartoon representations of the AlphaFold2-predicted and experimental KpsE structures. **h**, Coulomb surface potential of the octameric KpsE complex calculated in ChimeraX[55] (red: -10, blue:10 kcal/ mol[-1] $e$[-1]). **i**, Disordered density of the L2 loop visible at low contour levels. **j**, Comparison of KpsMT$_{E151Q}$ structures. Grey: the KpsMT$_{E151Q}$-KpsE complex; blue/salmon: KpsMT$_{E151Q}$ reconstituted into a Msp1e3d1 lipid nanodisc at about 3.8 Å resolution. The structures align with an RMSD of 1.1 Å between backbone atoms.

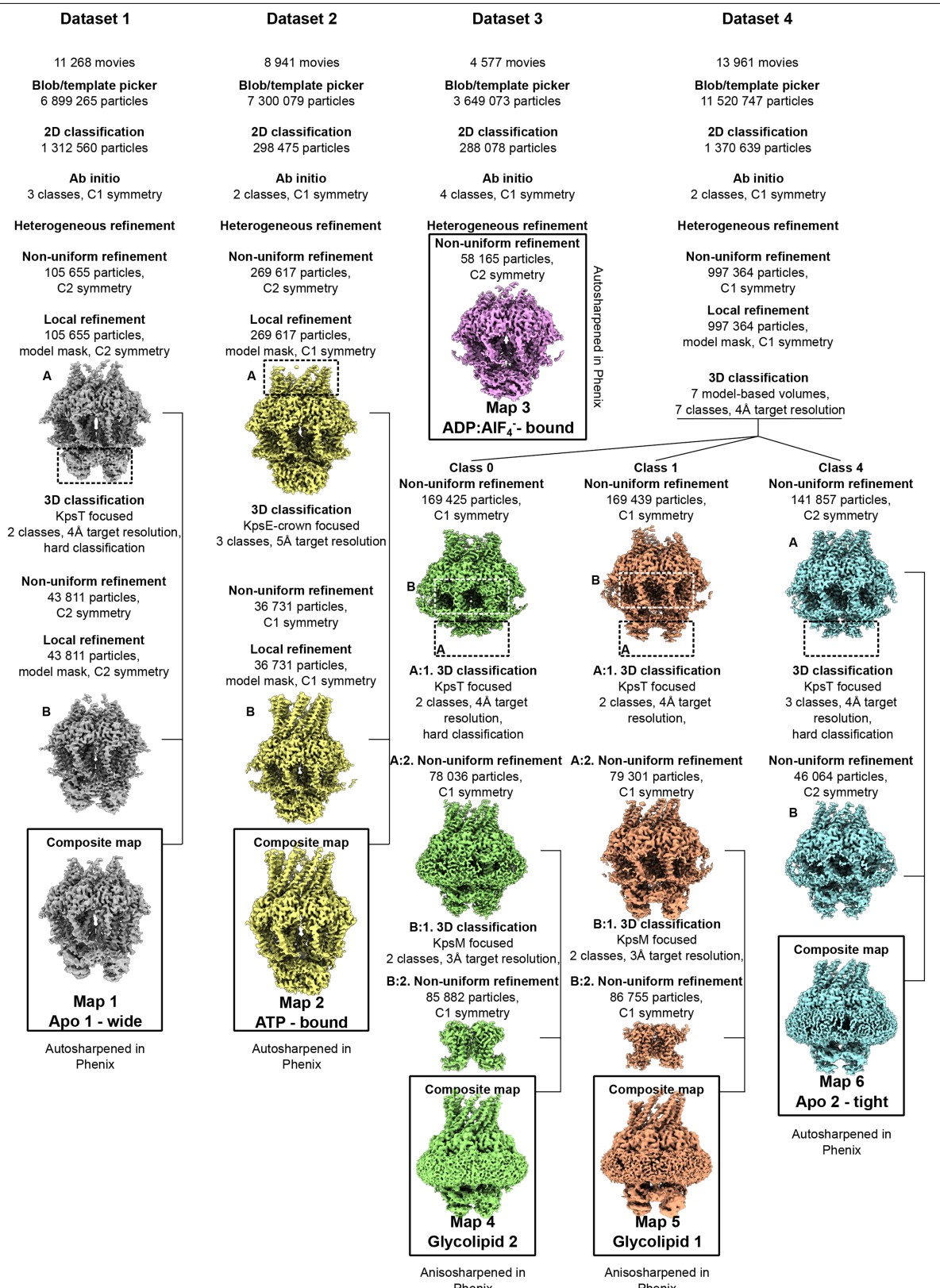

**Extended Data Fig. 3 | Cryo-EM data-processing workflows.** Datasets 1, 2 and 3 were processed in cryoSPARC v.3.3. Dataset 4 was processed in cryoSPARC v.4.0.

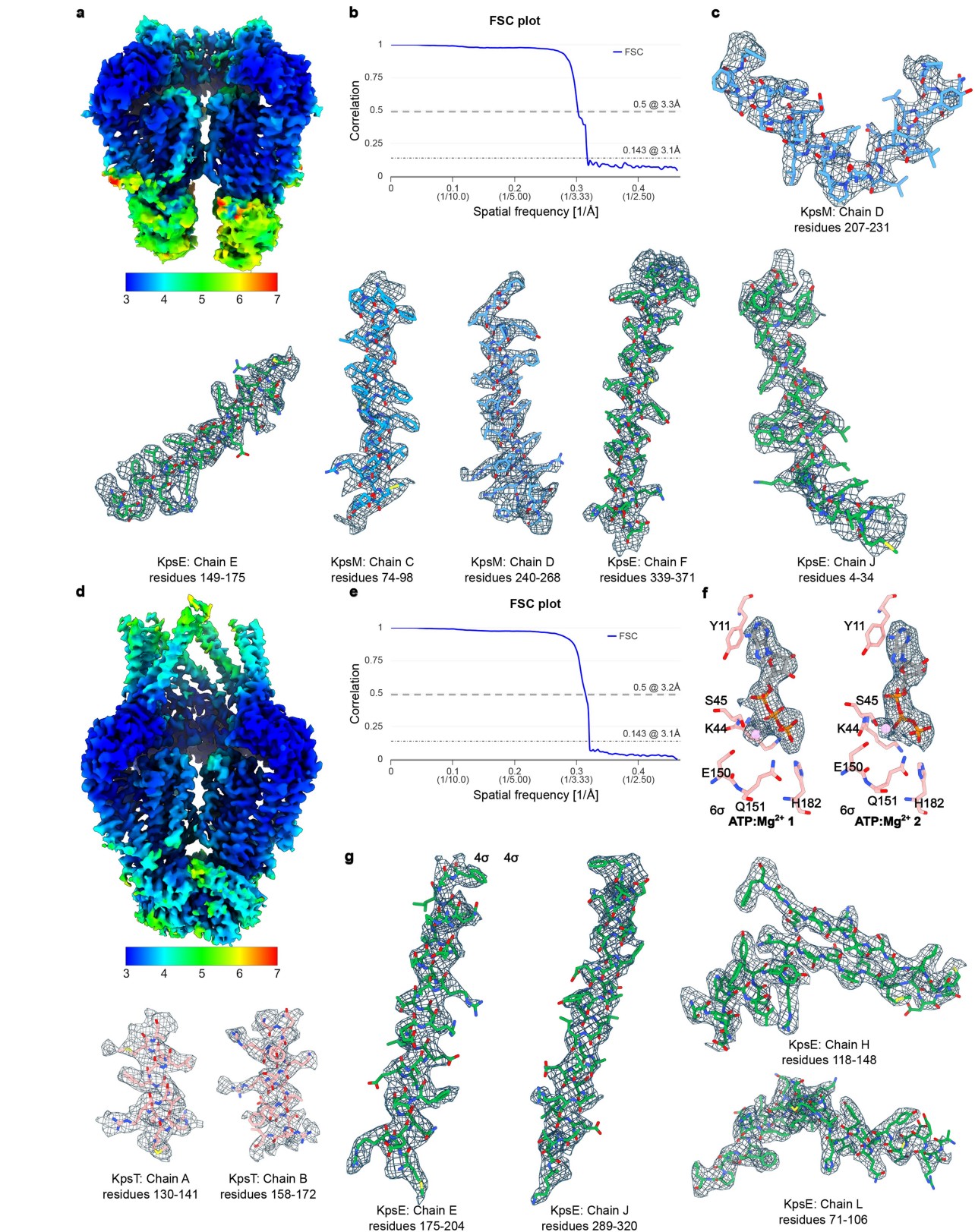

**Extended Data Fig. 4 | Local-resolution estimates and map quality examples for the Apo 1 and ATP-bound KpsMT–KpsE complexes. a**, Local-resolution estimate of map 1 (Apo 1 state). Two copies of KpsE were removed to show the KpsM volume. **b**, Gold-standard Fourier shell correlation (FSC) indicating the overall Map 1 resolution of 3.1 Å at FSC = 0.143. **c**, Examples of map 1 for the indicated segments. **d**, Local-resolution estimate of map 2 representing the ATP-bound state of KpsMT$_{E151Q}$-KpsE. Two copies of KpsE were removed to show the KpsM volume. **e**, Gold-standard Fourier shell correlation (FSC) indicating an overall map 2 resolution of 3.1 Å at FSC = 0.143. **f**, Cryo-EM maps of the ATP:Mg$^{2+}$ molecules. **g**, Examples of map 2 for the indicated segments. Cryo-EM densities in this figure are carved 2–3 Å from the models.

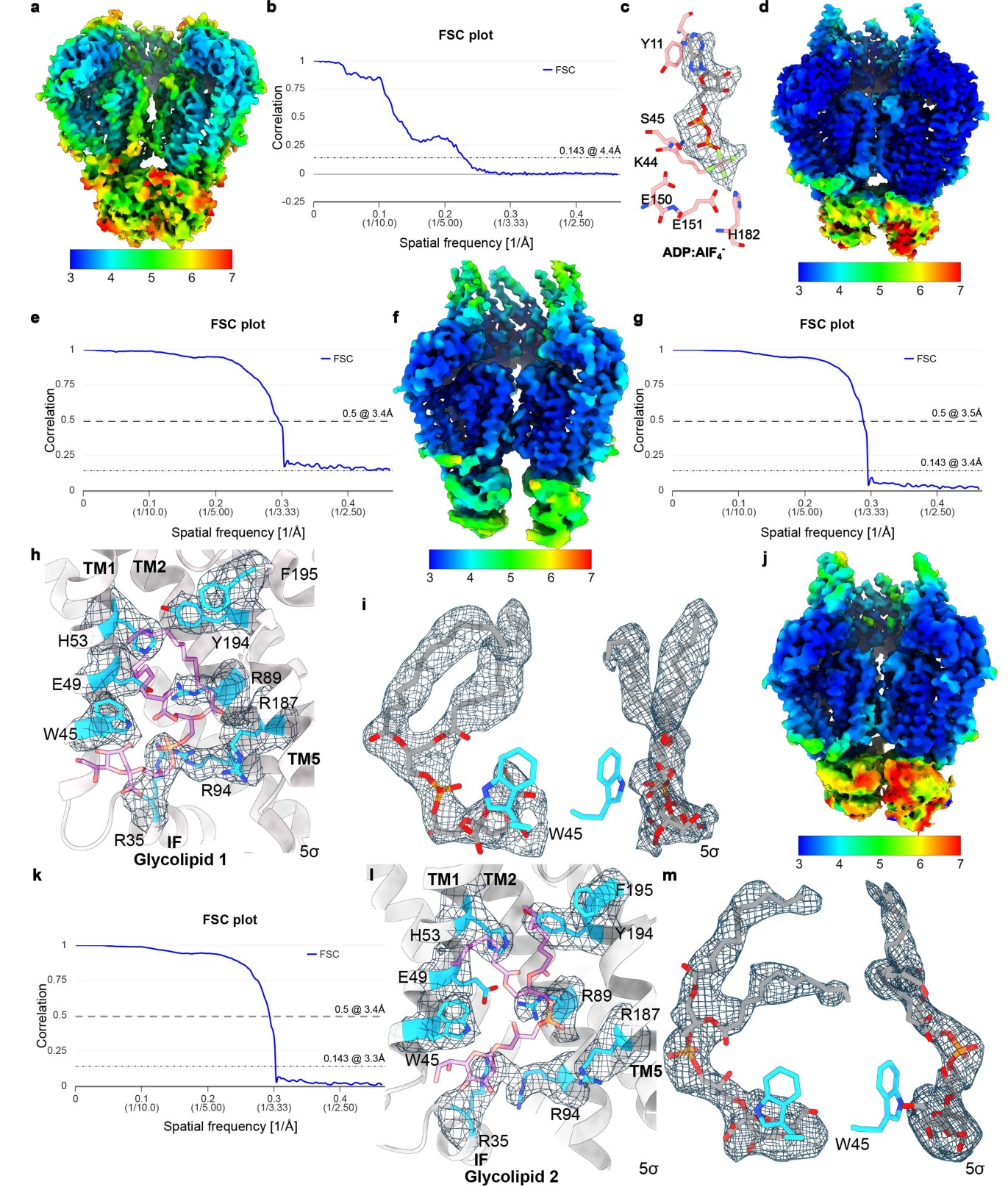

**Extended Data Fig. 5** | See next page for caption.

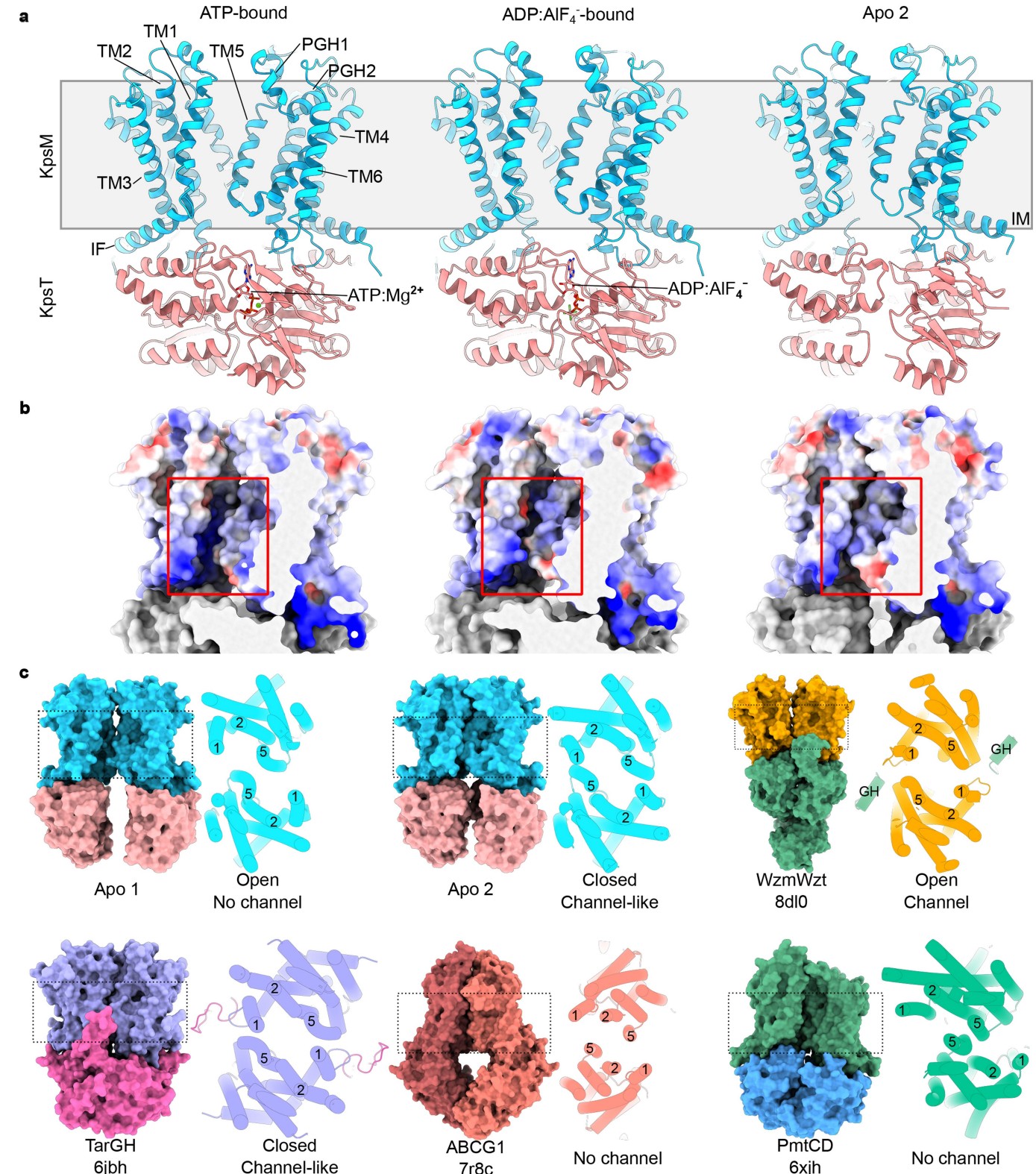

**Extended Data Fig. 6 | Comparison of the architecture of KpsMT with that of other ABC transporters. a**,**b**, KpsMT architecture in ATP-bound, ADP–AlF$_4^-$-bound and Apo 2 states (**a**) and corresponding Coulomb surface potentials (red: −10, blue:10 kcal/mol$^{-1}$ $e^{-1}$) (**b**). The polysaccharide canyon region is highlighted by a red box. TM: transmembrane helix, PGH: periplasmic gate helix, IF: interface helix. **c**, Comparison of the TM domain organization of KpsMT and other type-5 ABC transporters of known structure. GH: gate helix.

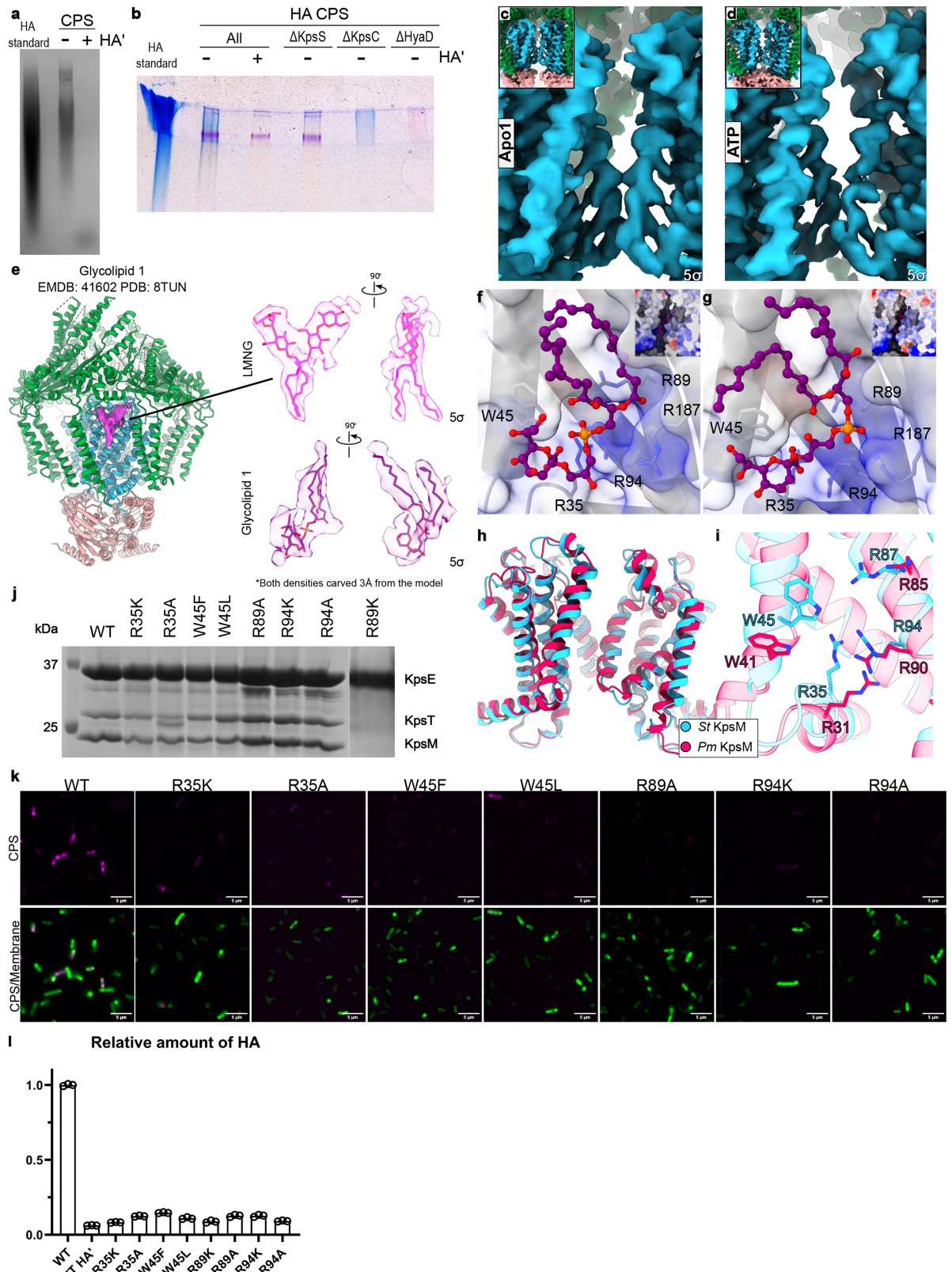

**Extended Data Fig. 7** | See next page for caption.

**Extended Data Fig. 7 | Glycolipid preparation and canyon mutagenesis.**
**a**, Agarose gel stained with Stains-All dye of recombinantly expressed and purified *Pm* CPS, run against a 300–500 kDa HA standard. HA': hyaluronidase treatment, 'All': expression of all *Pm* CPS components in *E. coli* C43ΔCPS1.
**b**, Polyacrylamide gel stained with Stains-All dye of recombinantly expressed *Pm* CPS, run against a 300–500 kDa HA standard. Components omitted during CPS expression are indicated above the lanes. HA part of CPS is stained blue, protein is stained pink. **c**,**d**, Cryo-EM map of Apo 1 and ATP-bound states presented at the same contour level and poses as glycolipid states 1 and 2 in Fig. 4a,d. The protein is coloured as in Fig. 2. **e**, Comparison of non-proteinaceous densities ascribed to an LMNG molecule and the putative Glycolipid 1. The molecules are observed in the Glycolipid 1 map (EMD-41602) and contoured at 5σ. The protein is coloured as in Fig. 2. **f**,**g**, Additional views of the glycolipid in states 1 and 2 inside the positively charged canyon. **h**, Overlay of the experimentally determined structure of *St* KpsM – Apo 1 with an AlphaFold2-predicted *Pm* KpsM model. **i**, Overlay of residues contributing to the polysaccharide canyon, underscoring the conservation of crucial residues. **j**, SDS-PAGE and Coomassie staining of purified KpsMT–KpsE complex mutants used in **h**. The R89K lane was added from another gel. **k**, In vivo encapsulation assay of KpsM mutants analysed by confocal fluorescence microscopy. CPS is labelled with the [Alexa647]SNAP–CBM70 probe, membranes are labelled with Cellbrite Fix 488. **l**, Relative amounts of surface-exposed HA detected on cells shown in j. HA was quantified using an ELISA-based HA detection kit. Results are normalized to the amount of HA present in WT cells. Experiments were done in triplicates and individual data points are shown.

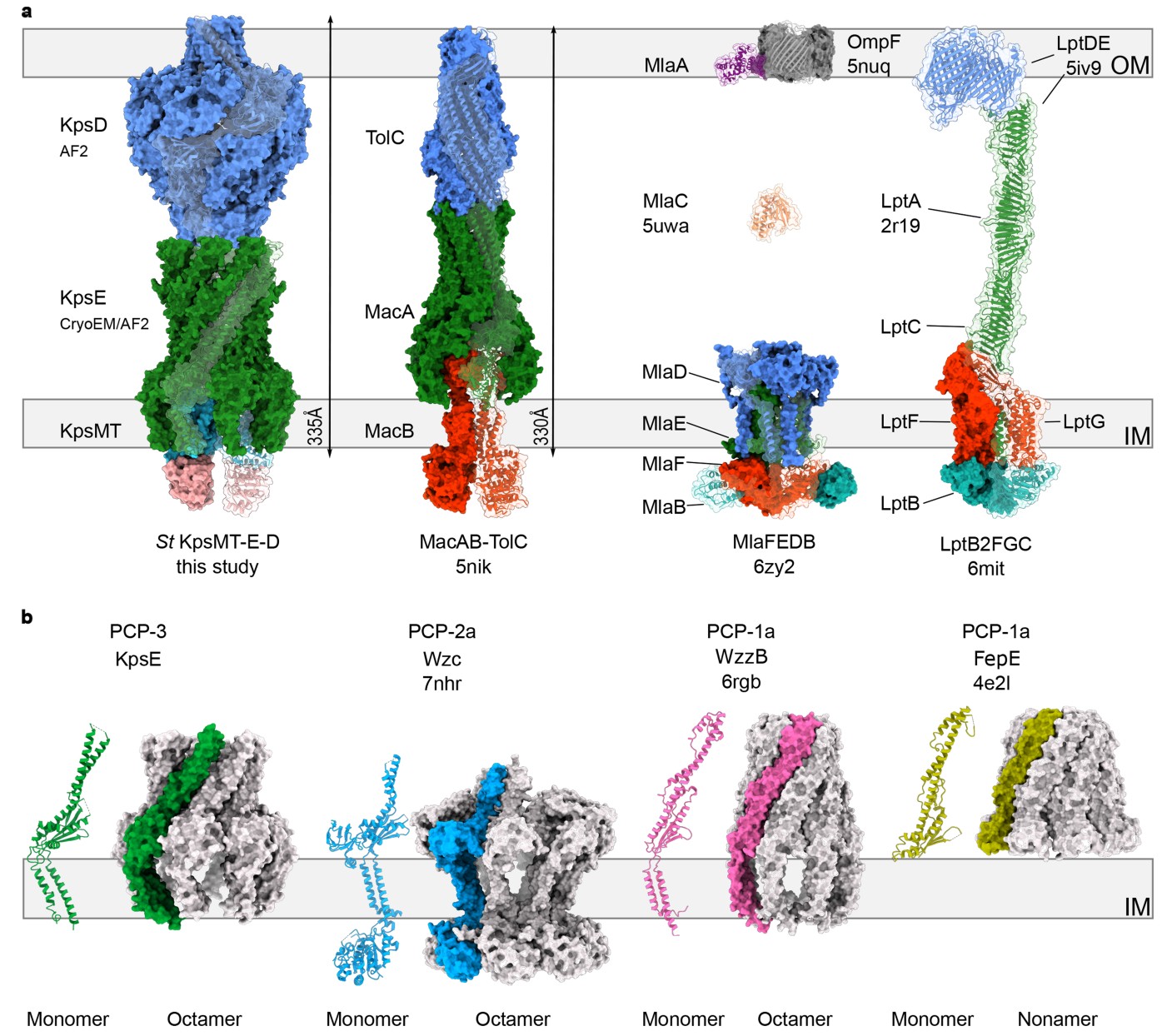

**Extended Data Fig. 8 | ABC-transporter-based trans-envelope secretion systems and known types of polysaccharide co-polymerases. a**, Organization of trans-envelope ABC-transporter-based secretion systems. The CPS system was manually assembled from AlphaFold2-predicted subcomplexes. **b**, Comparison of all three families of polysaccharide co-polymerases. PDB accession codes are listed below the protein names.

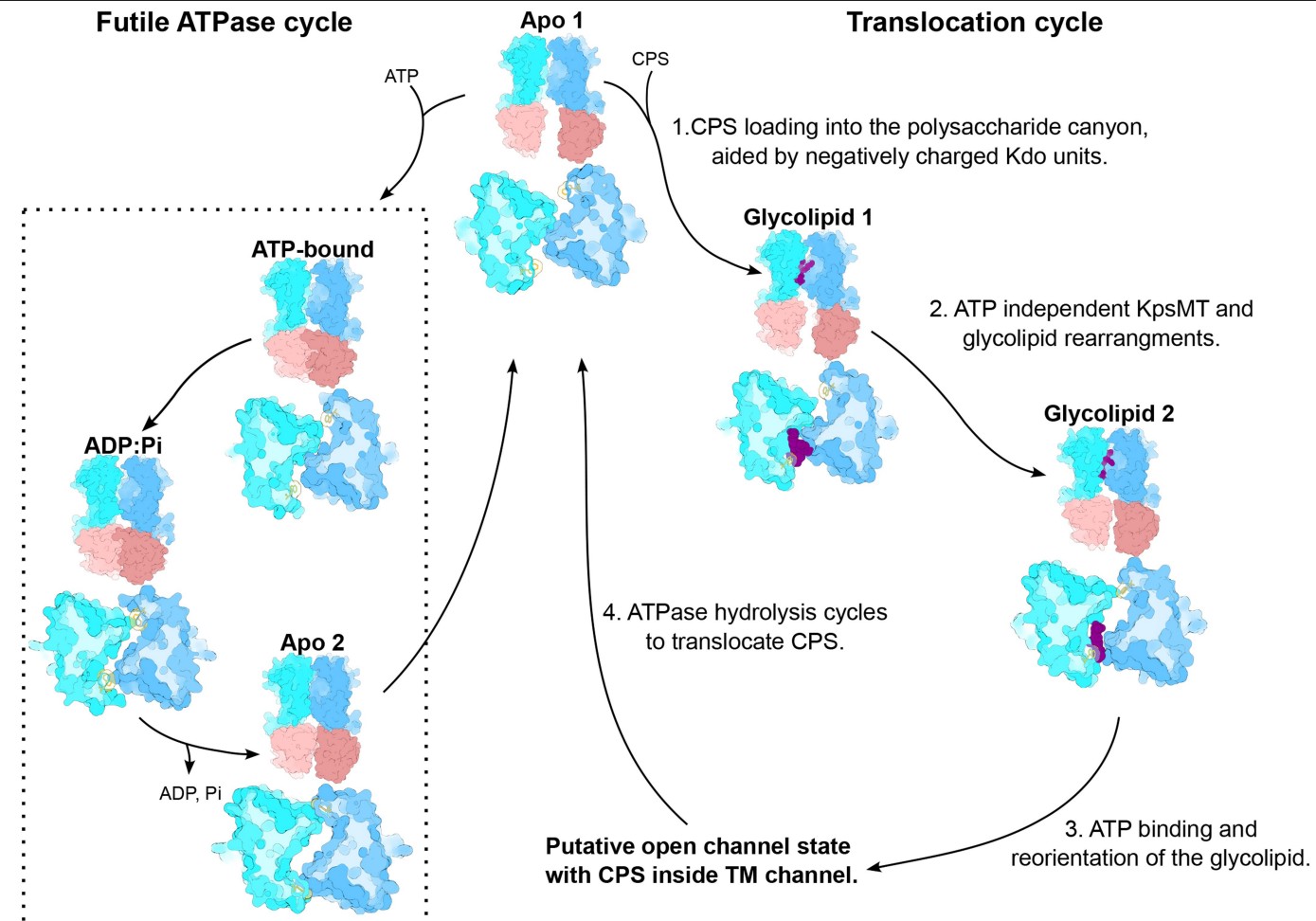

**Futile ATPase cycle**

**Apo 1**

ATP

CPS

1.CPS loading into the polysaccharide canyon, aided by negatively charged Kdo units.

**Translocation cycle**

**ATP-bound**

**ADP:Pi**

**Apo 2**

ADP, Pi

**Glycolipid 1**

2. ATP independent KpsMT and glycolipid rearrangments.

**Glycolipid 2**

4. ATPase hydrolysis cycles to translocate CPS.

3. ATP binding and reorientation of the glycolipid.

**Putative open channel state with CPS inside TM channel.**

**Extended Data Fig. 9 | KpsMT states captured in this study.** Possible conformational transitions during futile and substrate-loaded ATP hydrolysis cycles.

## Extended Data Table 1 | Cryo-EM data collection, refinement and validation statistics

| State | Apo 1 | ATP-bound | ADP:AlF$_4^-$ bound | Apo 2 | Glycolipid 1 | Glycolipid 2 |
|---|---|---|---|---|---|---|
| **Refined Subunits** | KpsEMT | KpsEMT | KpsEMT | KpsEM | KpsEM | KpsEM |
| **Refined Chains** | A,B,C,D,E,F,G,H,I,J,K,L | A,B,C,D,E,F,G,H,I,J,K,L | A,B,C,D,E,F,G,H,I,J,K,L | C,D,E,F,G,H,I,J,K,L | C,D,E,F,G,H,I,J,K,L | C,D,E,F,G,H,I,J,K,L |
| **Rigid Body Fitted Subunits** | - | - | - | KpsT | KpsT | KpsT |
| **Rigid Body Fitted Chains** | - | - | - | A,B | A,B | A,B |
| **PDB** | 8TSW | 8TSH | 8TSI | 8TSL | 8TUN | 8TT3 |
| **EMDB** | EMD-41601 | EMD-41592 | EMD-41593 | EMD-41595 | EMD-41626 | EMD-41602 |
| Data collection | | | | | | |
| Microscope | Titan Krios | | | | | |
| Camera | K3, GIF, 10eV slit | | | | | |
| Magnification | 81,000x | | | | | |
| Voltage (kV) | 300 | | | | | |
| Dose (e$^-$/Å$^2$) | 50 | | | | | |
| Defocus range (μm) | -1.8 to -1.0 | | | | | |
| Pixel size (Å) | 1.08 | | | | | |
| Data processing | | | | | | |
| Symmetry | C2 | C1 | C2 | C2 | C1 | C1 |
| Particles extracted # | 6 899 265 | 7 300 079 | 3 649 073 | 11 520 747 | | |
| Final Particles | 105 655 (Map A) 43 811 (Map B) | 269 617 (Map A) 36 731 (Map B) | 58,165 | 141 857 (Map A) 46 064 (Map B) | 79 301 (Map A) 86 755 (Map B) | 78 036 (Map A) 85 882 (Map B) |
| Map resolution (Å) | 3.1 (Map A) 3.3 (Map B) 3.1 (Composite) | 3.1 (Map A) 3.3 (Map B) 3.1 (Composite) | 4.4 | 3.3 (Map A) 3.5 (Map B) 3.4 (Composite) | 3.4 (Map A) 3.4 (Map B) 3.4 (Composite) | 3.3 (Map A) 3.3 (Map B) 3.3 (Composite) |
| FSC threshold | 0.143 | 0.143 | 0.143 | 0.5 | 0.143 | 0.143 |
| Refinement | | | | | | |
| Initial model used | AF2 | Apo 1 | | | | |
| Model resolution (Å) | 3.4 | 3.3 | 4.0 | 3.4 | 3.7 | 3.6 |
| Non-hydrogen atoms | 20 318 | 24 235 | 20 242 | 21 382 | 22 037 | 22 741 |
| Protein residues | 2506 | 3001 | 2488 | 2646 | 2726 | 2820 |
| Ligands | 0 | 2: ATP 2: Mg$^{2+}$ | 2: ADP:AlF$_4^-$ | 0 | 1: PGK | 1: PGK |
| B Factors (Å$^2$) | | | | | | |
| Protein | 84.72 | 86.77 | 39.14 | 104.21 | 144.56 | 137.93 |
| Ligands | - | 71.34 | 76.52 | - | 137.36 | 118.76 |
| RMS deviations | | | | | | |
| Bond lenghts | 0.003 | 0.003 | 0.003 | 0.004 | 0.005 | 0.004 |
| Bond angles | 0.617 | 0.572 | 0.671 | 0.670 | 0.826 | 0.734 |
| Validation | | | | | | |
| MolProbity score | 1.58 | 1.46 | 1.71 | 1.55 | 1.7 | 1.56 |
| Clashscore | 6.79 | 6.77 | 8.1 | 6.88 | 9.05 | 7.27 |
| Rotamer outliers (%) | 0.27 | 0.19 | 0.68 | 0.48 | 0.97 | 0.37 |
| Ramachandran plot | | | | | | |
| Favored | 96.73 | 97.59 | 96.09 | 97.03 | 96.59 | 97.07 |
| Allowed | 3.27 | 2.41 | 3.91 | 2.97 | 3.42 | 2.93 |
| Outliers | 0.0 | 0.0 | 0.0 | 0.0 | 0.0 | 0.0 |

KJ9: phosphatidylglycerol-linked Kdo.

# Reporting Summary

## Statistics

For all statistical analyses, confirm that the following items are present in the figure legend, table legend, main text, or Methods section.

| n/a | Confirmed | |
|---|---|---|
| ☐ | ☒ | The exact sample size (*n*) for each experimental group/condition, given as a discrete number and unit of measurement |
| ☐ | ☒ | A statement on whether measurements were taken from distinct samples or whether the same sample was measured repeatedly |
| ☒ | ☐ | The statistical test(s) used AND whether they are one- or two-sided<br>*Only common tests should be described solely by name; describe more complex techniques in the Methods section.* |
| ☒ | ☐ | A description of all covariates tested |
| ☒ | ☐ | A description of any assumptions or corrections, such as tests of normality and adjustment for multiple comparisons |
| ☒ | ☐ | A full description of the statistical parameters including central tendency (e.g. means) or other basic estimates (e.g. regression coefficient) AND variation (e.g. standard deviation) or associated estimates of uncertainty (e.g. confidence intervals) |
| ☒ | ☐ | For null hypothesis testing, the test statistic (e.g. *F*, *t*, *r*) with confidence intervals, effect sizes, degrees of freedom and *P* value noted<br>*Give P values as exact values whenever suitable.* |
| ☒ | ☐ | For Bayesian analysis, information on the choice of priors and Markov chain Monte Carlo settings |
| ☒ | ☐ | For hierarchical and complex designs, identification of the appropriate level for tests and full reporting of outcomes |
| ☒ | ☐ | Estimates of effect sizes (e.g. Cohen's *d*, Pearson's *r*), indicating how they were calculated |

*Our web collection on statistics for biologists contains articles on many of the points above.*

## Software and code

Policy information about availability of computer code

| Data collection | EPU, Imspector, Zeiss ZEN |
|---|---|
| Data analysis | CryoSPARC, Phenix, Coot, ChimeraX, ParaView, GraphPad Prism, ImageJ |

For manuscripts utilizing custom algorithms or software that are central to the research but not yet described in published literature, software must be made available to editors and reviewers. We strongly encourage code deposition in a community repository (e.g. GitHub). See the Nature Portfolio guidelines for submitting code & software for further information.

## Data

Policy information about availability of data

All manuscripts must include a data availability statement. This statement should provide the following information, where applicable:

- Accession codes, unique identifiers, or web links for publicly available datasets
- A description of any restrictions on data availability
- For clinical datasets or third party data, please ensure that the statement adheres to our policy

*Provide your data availability statement here.*

# Research involving human participants, their data, or biological material

Policy information about studies with human participants or human data. See also policy information about sex, gender (identity/presentation), and sexual orientation and race, ethnicity and racism.

| | |
|---|---|
| Reporting on sex and gender | NA |
| Reporting on race, ethnicity, or other socially relevant groupings | NA |
| Population characteristics | NA |
| Recruitment | NA |
| Ethics oversight | NA |

Note that full information on the approval of the study protocol must also be provided in the manuscript.

# Field-specific reporting

Please select the one below that is the best fit for your research. If you are not sure, read the appropriate sections before making your selection.

☒ Life sciences    ☐ Behavioural & social sciences    ☐ Ecological, evolutionary & environmental sciences

For a reference copy of the document with all sections, see nature.com/documents/nr-reporting-summary-flat.pdf

# Life sciences study design

All studies must disclose on these points even when the disclosure is negative.

| | |
|---|---|
| Sample size | Each activity assay was performed in technical triplicates or quadruplicates. Each cryoEM dataset was collected for a single sample. |
| Data exclusions | none |
| Replication | Multiple cryo EM datasets were collected and the highest quality datasets were used for processing and model building. Multiple fluorescence microscopy images were taken for each sample. For each Minflux experiment at least three independently prepared measurements were conducted. |
| Randomization | none |
| Blinding | none |

# Reporting for specific materials, systems and methods

We require information from authors about some types of materials, experimental systems and methods used in many studies. Here, indicate whether each material, system or method listed is relevant to your study. If you are not sure if a list item applies to your research, read the appropriate section before selecting a response.

## Materials & experimental systems

| n/a | Involved in the study |
|---|---|
| ☐ | ☒ Antibodies |
| ☒ | ☐ Eukaryotic cell lines |
| ☒ | ☐ Palaeontology and archaeology |
| ☒ | ☐ Animals and other organisms |
| ☒ | ☐ Clinical data |
| ☒ | ☐ Dual use research of concern |
| ☒ | ☐ Plants |

## Methods

| n/a | Involved in the study |
|---|---|
| ☒ | ☐ ChIP-seq |
| ☒ | ☐ Flow cytometry |
| ☒ | ☐ MRI-based neuroimaging |

## Antibodies

| | |
|---|---|
| Antibodies used | Commercially available primary anti-His, -FLAG, -Strep, and -Myc antibodies, as well as secondary anti-Mouse (Rockland) antibodies for Western Blots. |

Validation

By manufacturer.

