## [Peer Review File · Nature]

Manuscript Title: Molecular insights into capsular polysaccharide secretion

Reviewer Comments & Author Rebuttals

Reviewer Reports on the Initial Version:

Referees' comments:

Referee #1 (Remarks to the Author):

Kuklewicz and Zimmer report the structural investigation of KpsMT-KpsE, part of an ABC transport complex responsible for the export of capsular polysaccharides (CPS) across the bacterial cell envelope. The authors show that KpsMT-KpsE is capable of exporting *P. multocida* CPS produced in *E. coli* cells, establishing an assay for structure-function studies. Using cryo EM, the authors determine structures of KpsMT-KpsE in several different states of the transport cycle. These structures reveal how the largely-periplasmic KpsE forms a “cage” around the KpsMT ABC transporter in the inner membrane, and the presence of a “canyon” at the interface between the two KpsM subunits where CPS substrates are observed bound in the EM density maps.

Overall, the structures appear to be of high quality and modeled with rigor. The density for the CPS substrate is convincing and interpreted/discussed with clarity about which parts are clear and which parts are ambiguous. The structures have been thoughtfully analyzed and discussed. I think this is a very solid manuscript and I have relatively minor comments and suggestions for revision.

MAJOR COMMENTS:

None

MINOR COMMENTS:

In 89-91 and Fig 1e: The polar localization of CPS is surprising to me; based on EM images, I thought capsule was roughly evenly distributed around the surface of the cell. Am I missing something? Is this imaging done after only a short pulse of expression, such that only sites of active CPS translocation have mature capsule and therefore only the poles stain?

In 200-208: Are there any parallels between these KpsE TM interactions with the ABC transporter and other type V/VI/VIII TMDs, like LptC's TM with LptFG or MlaD's TM with MlaE. It would be interesting if any of these sites were shared and defined a conserved "docking site" for associated periplasmic subunits.

In 221-222: "minimal contacts between TM helices 5": I interpreted these words as meaning that the

TM5s essentially don't interact, and that the dimer interface is formed by other structural elements, but this is not what the figures show. In retrospect, perhaps the wording here is unclear, as if I understand correctly, the ONLY contacts at the KpsM interface are mediated by TM5, but those contacts are not particularly extensive/tight.

In 222-223: My first thought reading this and looking at Fig. 3a was of ABCA1, not ABCG. Just wanted to confirm this wasn't a typo, but perhaps ABCG is actually more similar.

Add labels for KpsM and T to Fig 3a?

Fig 3b: While the upper panels showing the TM helices above are clear, the conformational changes shown in surface representation are a bit hard for me to follow. Perhaps these transitions could be shown another way? A movie morphing between states would definitely help. In addition/alternatively, another option could be to show the 4 states separately, side-by-side, and color a few key residues on the surface as markers of the movement (e.g., two residues in KpsM that are in close contact in the ADP:AlF₄ structure but far apart in Apo1 colored red so the reader can see the distance between them change).

ED Fig 7a: I found the "All" label confusing here. When a reader is first directed to this figure, it isn't clear that in subsequent figures CPS isolation will be attempted from strains that don't have "All" of the key gene products. I think using the "HA CPS" label as in ED Fig 7b would be more clear and avoid my confusion

In 271-275: The assignment of the density in the canyon as CPS rests in large part on the appearance of this density in samples where CPS was added and its absence when CPS was not added. To support this, I think the authors should add a panel like Fig 4a/c (similar contour level, highlighting non-protein density in purple) based on the most similar structure from Fig. 3, showing what ordered density is present in the canyon in the absence of substrate.

Fig 4c: Indicate additional unmodeled density with arrow or different color?

ED Fig 5, h and l: I am a little confused as to why there isn't density on the CPS molecules in these figures. Is the map "carved" around the labeled residues? I think that should be stated in the legend (as well as in other carved figures, like i and m).

In 316-317, ED Fig 7g: I think the data for R89K should be shown. Perfectly fine that this complex is unstable, but if stated in the text it should be formally shown (e.g., a gel lane lacking 1 or more bands).

In 309-315, ED Fig. 7h: Can this assay be quantified in some way (e.g., mean fluorescence per bacterium? Percent bacteria scored "positive" based on some cutoff?). I like that the authors are showing representative images, but quantification and data on reproducibility, number of biological replicates, etc. is currently missing.

In 325: Here the authors refer to KpsE as the "'membrane fusion' component". Is there any structural similarity to MFPs from the Acr or Mac systems, for example? That seems like an interesting area for comparison/contrast. Also, there wasn't much said about how KpsE compares to previously determined structures, and/or structurally what distinguishes PCP-3 proteins like KpsE from PCP-1 and PCP-2. Based on ED Fig 8b, it looks like they are probably all structurally related to one another?

ED Table 1: In final table, please include PDB codes, EMDB codes, and EMPIAR codes for easy reference

Please deposit your raw EM data in EMPIAR so it will be available to the community for years to come!

Referee #2 (Remarks to the Author):

In this interesting work the process of capsular polysaccharide (CPS) secretion through an ABC transporter in gram-negative bacteria complex was studied. The secretion process was described on the structural and molecular level employing techniques like protein crystallization, cryo EM, Minflux superresolution fluorescence microscopy, biochemical characterization and AlphaFold modelling. Minflux data are shown in Fig. 1 f and in Extended Data Figure 1 g-j. The Minflux data clearly contribute to the key findings of the work by visualizing in high resolution the surface organization of CPS and KpsD, the latter being the outer membrane pore of the ABC transporter. Minflux allowed to visualize the various CPS formations as well as helped to estimate the total number of KpsD clusters on the surface of single bacterial cells. It also appeared to document spatially corresponding KpsD and CPS clusters. On the technical side, 2-colour Minflux in 2-D mode was employed and the pictures obtained are of high quality.

One caveat is that 2-D Minflux does not provide high resolution in the z-axis (around 600 nm depending on the pinhole size), therefore, the spatial alignment of the CPS and KpsD clusters against each other cannot be visualized at high resolution. In case the authors have the possibility to perform 3-D Minflux with their instrument setup this would improve depiction of the 3-D aspects of CPS and KpsD organization. We acknowledge that Minflux microscopes and setups are still rare and the setup used here might not allow to perform 3-D mode.

Minor points:

Line 125: It would be useful to mention here why the exact number of KpsD clusters cannot be determined with the current labeling strategy?

Fig. 1 f; Extended Fig. 1 g, i: The number of KpsD clusters seems to be lower in the 2-color Minflux experiments. Does this just reflect biological variation or is it a technical artifact due to different on-/off-switching behavior of the two dyes in the GLOX buffer, whereby one dye might "overshine" the other ?

Referee #3 (Remarks to the Author):

Kuklewicz and Zimmer present here a comprehensive structural and biochemical analysis of capsular polysaccharide secretion as it is found in many Gram-negative bacteria.

At the core of the machinery is a Type V ABC transporter called KpsMT, which recognizes the phosphatidyl-choline-attached polysaccharide across the inner membrane at the expense of ATP hydrolysis in a process that is reminiscent of the O-antigen transport system WzmWzt. KpsMT is assisted with the membrane protein KpsE, which as shown in this paper forms a large channel surrounding a single KpsMT complex and protrudes far into the periplasm to form a bridge to the outer membrane channel KpsD. The entire transport machinery spans both membranes and is required for the surface decoration of Gram-negative cells with capsular polysaccharides, which can in fact be composed of chains of several hundreds of sugar units.

The story stands on two strong pillars, namely strain engineering and light/fluorescence microscopy analyses of the machinery in situ and an extensive single particle cryo-EM analysis.

The authors could convincingly show which proteins of the Kps operon are necessary for polysaccharide polymerization. The authors could further demonstrate that the outer membrane channel KpsD is dispensable for transporting the lipid-linked polysaccharide across the inner membrane (as shown with spheroplasts), but required for the sugar chains arriving at the cell surface. What remains unclear, however, is whether the KpsE adaptor protein is required for transport of glycolipids across the inner membrane.

At the level of the cryo-EM analysis, the authors for the first time showed the carousel-like structure of KpsE in complex with the KpsMT transporter. Further, they trapped KpsMT in several conformational states. Noteworthy (and certainly also a bit lucky), the authors apparently achieved to visualize the native substrate (which they purified themselves) in two binding poses interacting apo KpsMT-KpsE complex, thereby showing how KpsMT interacts with the lipid moiety and the core sugars immediately following the lipid. While the sugars were not highly resolved, the structural details were sufficient to locate three arginine residues present in a what the authors call an electropositive canyon within KpsMT, which interact with the phosphate groups of the lipid and which they could show in convincing mutational experiments involving very mild substitutions (e.g. R to K or W to F) to be of key importance for capsular polysaccharide secretion. Visualization of the glycolipids provides strong evidence for a model wherein the lipid moiety together with conserved core sugars directly attached to it are recognized first by the ABC transporter and brought on a journey towards the periplasm and finally cell surface. The onward journey, and in particular the exit gate within the KpsMT transporter through which the sugar chain has to finally be ratcheted through in a processive manner requiring ATP hydrolysis remains unfortunately an elusive process lacking molecular elucidation. Yet, the in situ experiments clearly show that it does indeed happen.

This is an excellent paper, as it offers novel insights that are solidly supported by experimental evidence. The authors also formulate more carefully at instances where the experimental data do provide only partial or suboptimal answers (e.g. quality of the maps to locate the sugar moieties of the substrate). What makes this paper special is the way the author combined high quality fluorescence microscopy imaging and functional analyses using an engineered E. coli strain with state-of-the-art cryo-EM analysis. The paper is a joy to read and also the figures, though some improvements here and there might improve clarity, are appealing and clear.

Point to address

- 1) I did not see experiments showing that KpsE is needed for glycolipid translocation of KpsMT across the inner membrane. The spheroblasts assays shown in Fig. 1 could be used to clarify this point. This question also boils down to the more fundamental question on the molecular role of KpsE: is it more than just a periplasmic bridge linking the transporter to the channel (also in analogy to AcrA for example in tripartite efflux pumps in *E. coli*, which in fact has mostly this bridging/tunnelling role).
- 2) It is unclear from the descriptions in the paper how the tripartite complex of KpsMT-KpsE was formed? Did the authors express all three components at the same time in the cells and co-purified them? Or were they separately purified and finally mixed together? The materials and methods do not specify this clearly.
- 3) Did the authors purify the KpsMT ABC transporter alone and tried to determine its structure? It would have been interesting to see whether KpsE modulates the structure and thus function of KpsMT.
- 4) The authors show basal ATPase activities of KpsMT-KpsE. Did the authors try to look at stimulated ATPase activity in the presence of the glycolipid? Or by having KpsE present or absent?
- 5) The authors used a cross-linked mutant of KspE in functional and structural experiments. I was wondering if they performed *in vivo* experiments (fluorescent microscopy) with WT KspE and noticed any differences with the cross-linked mutant in terms of the speed and extent of hyaluronan (HA) export.
- 6) The authors mention that in all cryo-EM structures of KspMT-KspE, the KspE cage looks the same. One may argue that identical conformations were observed because KspE was cross-linked?
- 7) The authors observe additional cryo-EM densities in protein sample where they added purified substrate and interpret them as glycolipids. Can the authors rule out that this non-proteinaceous density actually represents a molecule of the detergent LMNG used for protein purification? The shape of LMNG resembles the shape of a lipid, at least at the resolution of the cryo-EM density in this region and highly ordered LMNG binding sites are not uncommon.

Author Rebuttals to Initial Comments:

We would like to thank the reviewers for the thorough review of our manuscript. We appreciate their support and enthusiasm about our work on CPS secretion.

MAJOR COMMENTS:

None

MINOR COMMENTS:

In 89-91 and Fig 1e: The polar localization of CPS is surprising to me; based on EM images, I thought capsule was roughly evenly distributed around the surface of the cell. Am I missing something? Is this imaging done after only a short pulse of expression, such that only sites of active CPS translocation have mature capsule and therefore only the poles stain?

>We believe that our engineered *E. coli* cells do not produce a 'mature capsule', perhaps due to instability of the polycistronic expression plasmids or inadequate expression levels of some of the components. Polar localization of CPS has also been observed in *E. coli* for a natively expressed group 2 capsular polysaccharide using immunofluorescence with a K5-specific monoclonal antibody (McNulty 2006). In this work, it was observed that encapsulation starts at the poles and then spreads over entire cell over time. In our engineered system, most of the cells never reach full encapsulation and CPS localization is limited to cell poles and 'secretion hotspots. We state in the text that the engineered capsule does not reflect a 'mature capsule' to emphasize this point.

In 200-208: Are there any parallels between these KpsE TM interactions with the ABC transporter and other type V/VI/VIII TMDs, like LptC's TM with LptFG or MlaD's TM with MlaE. It would be interesting if any of these sites were shared and defined a conserved "docking site" for associated periplasmic subunits.

> We looked but did not identify a consensus 'docking motif' that mediates interactions with additional components, as the reviewer suggests. Indeed, it would be great to be able to predict these docking sites.

In 221-222: "minimal contacts between TM helices 5": I interpreted these words as meaning that the TM5s essentially don't interact, and that the dimer interface is formed by other structural elements, but this is not what the figures show. In retrospect, perhaps the wording here is unclear, as if I understand correctly, the ONLY contacts at the KpsM interface are mediated by TM5, but those contacts are not particularly extensive/tight.

> Thank you for pointing this out. The text has been revised.

In 222-223: My first thought reading this and looking at Fig. 3a was of ABCA1, not ABCG. Just wanted to confirm this wasn't a typo, but perhaps ABCG is actually more similar.

> Based on a Foldseek search conducted in PDB100 and using the KpsMT Apo1 state as a reference, ABCG1 has a higher score compared to ABCA1. Therefore, we included ABCG1 in our selection of type V ABC transporters in ED Fig 6c.

Add labels for KpsM and T to Fig 3a?

> Done as requested.

Fig 3b: While the upper panels showing the TM helices above are clear, the conformational changes shown in surface representation are a bit hard for me to follow. Perhaps these transitions could be shown another way? A movie morphing between states would definitely help. In addition/alternatively, another option could be to show the 4 states separately, side-by-side, and color a few key residues on the surface as markers of the movement (e.g., two residues in KpsM that are in close contact in the ADP:AIF4 structure but far apart in Apo1 colored red so the reader can see the distance between them change).

> Thank you for the suggestion. We included a morph in the supplemental materials (see SI movie 3) to stress conformational transitions of the ABC transporter.

ED Fig 7a: I found the "All" label confusing here. When a reader is first directed to this figure, it isn't clear that in subsequent figures CPS isolation will be attempted from strains that don't have "All" of the key gene products. I think using the "HA CPS" label as in ED Fig 7b would be more clear and avoid my confusion

> We have changed the label accordingly.

In 271-275: The assignment of the density in the canyon as CPS rests in large part on the appearance of this density in samples where CPS was added and its absence when CPS was not added. To support this, I think the authors should add a panel like Fig 4a/c (similar contour level, highlighting non-protein density in purple) based on the most similar structure from Fig. 3, showing what ordered density is present in the canyon in the absence of substrate.

> The requested panels are now shown as ED Fig. 7c and d. We do not find any additional density inside the canyon. Please also see the response to point #7 of reviewer #3.

Fig 4c: Indicate additional unmodeled density with arrow or different color?

> We have modified Fig. 4 panels c and f to improve the clarity of this Figure. The putative polysaccharide density is better visualized from a different viewing direction.

ED Fig 5, h and i: I am a little confused as to why there isn't density on the CPS molecules in these figures. Is the map "carved" around the labeled residues? I think that should be stated in the legend (as well as in other carved figures, like i and m).

> The lipid densities were omitted from these views to improve the visualization of the coordinating side chains. The corresponding glycolipid densities from the same map

and at the same contour level are shown in the separate panels (i and m). We have revised the caption to avoid confusion.

In 316-317, ED Fig 7g: I think the data for R89K should be shown. Perfectly fine that this complex is unstable, but if stated in the text it should be formally shown (e.g., a gel lane lacking 1 or more bands).

> This is now included.

In 309-315, ED Fig. 7h: Can this assay be quantified in some way (e.g., mean fluorescence per bacterium? Percent bacteria scored "positive" based on some cutoff?). I like that the authors are showing representative images, but quantification and data on reproducibility, number of biological replicates, etc. is currently missing.

> Thank you for the suggestion. To address this point, we quantified the total amount of surface-exposed HA using a commercially available ELISA-based HA detection kit (Echelon Biosciences). The results agree with the confocal imaging and are now included as ED Fig. 7k.

In 325: Here the authors refer to KpsE as the "'membrane fusion' component". Is there any structural similarity to MFPs from the Acr or Mac systems, for example? That seems like an interesting area for comparison/contrast. Also, there wasn't much said about how KpsE compares to previously determined structures, and/or structurally what distinguishes PCP-3 proteins like KpsE from PCP-1 and PCP-2. Based on ED Fig 8b, it looks like they are probably all structurally related to one another?

> To avoid confusion, we rephrased the sentence to "... is fully encircled by the associated periplasmic subunit KpsE." Despite some common features, such as extended helical 'hairpins', the AcrA and MacA architectures are quite different in the membrane proximal regions. Considering length limitations, a proper structural and functional comparison may be better suited for a review article in the future.

ED Table 1: In final table, please include PDB codes, EMDB codes, and EMPIAR codes for easy reference

> We updated the ED Table 1 with the PDB and EMDB codes and are in the process of uploading movies to EMPIAR.

Referee #2 (Remarks to the Author):

One caveat is that 2-D Minflux does not provide high resolution in the z-axis (around 600 nm depending on the pinhole size), therefore, the spatial alignment of the CPS and KpsD clusters against each other cannot be visualized at high resolution. In case the authors have the possibility to perform 3-D Minflux with their instrument setup this would improve depiction of the 3-D aspects of CPS and KpsD organization. We acknowledge that Minflux microscopes and setups are still rare and the setup used here might not allow to perform 3-D mode.

> We have included two supplementary movies presenting 3D Minflux datasets. In the first movie (SI movie 1), we show a cell with a volcano-like CPS formation. The second movie (SI movie 2) displays the results of 2-color 3D Minflux, revealing KpsD clusters as well as CPS secretion hotspots. It's important to note that finding a suitable target cell for 2-color analysis in 3D is challenging due to:

- 1) Not all cells in the prepared samples show CPS and KpsD labeling. At the confocal level, when selecting a specific cell for Minflux analysis, we cannot determine whether both labels are present or not since both fluorophores emit in the red channel.
- 2) Related to the point above, CPS secreting KpsD complexes may be inaccessible to the KpsD probe due to crowding and/or electrostatic repulsion (HA and Alexa680 are both negatively charged). This will be addressed in the future using a different labeling strategy.

Minor points:

Line 125: It would be useful to mention here why the exact number of KpsD clusters cannot be determined with the current labeling strategy?

> In the absence of a proper reference and considering variability in labeling, we are uncertain about the size of a localization cloud corresponding to a single functional KpsD oligomer. Larger patches may correspond to KpsD patches or single secretion systems. Partially assembled subcomplexes or even monomeric subunits are likely present as well. In addition, in super-resolution microscopy, the overall fluorescence intensity or density of localizations does not scale linearly with the fluorophore abundance as other factors, such as dark states and bleaching, can affect the detectability. The text has been revised to reflect this uncertainty.

Fig. 1 f; Extended Fig. 1 g, i: The number of KpsD clusters seems to be lower in the 2-color Minflux experiments. Does this just reflect biological variation or is it a technical artifact due to different on-/off-switching behavior of the two dyes in the GLOX buffer, whereby one dye might “overshine” the other?

> As mentioned above, we suspect that CPS could hinder KpsD labeling. Modifying the labeling strategy—such as employing non-natural amino acids for direct KpsD labeling—might offer a potential solution. In addition, biological variability introduces another layer of complexity, as illustrated in the additional examples of the 2-color Minflux analysis shown in the Supplementary Information (SI).

Referee #3 (Remarks to the Author):

Point to address

1) I did not see experiments showing that KpsE is needed for glycolipid translocation of KpsMT across the inner membrane. The spheroblasts assays shown in Fig. 1 could be used to clarify this point. This question also boils down to the more fundamental question on the molecular role of KpsE: is it more than just a periplasmic bridge linking

the transporter to the channel (also in analogy to AcrA for example in tripartite efflux pumps in *E. coli*, which in fact has mostly this bridging/tunnelling role).

> Thank you for the suggestion. We have added a spheroplast experiment in Fig. 1g that addresses this point. Indeed, in our experimental system, KpsMT is functional in the absence of KpsE and translocates the glycolipid across the inner membrane and to the periplasm. Whether this is due to overexpression of the CPS operon or of physiological relevance remains to be addressed. With that in mind, we hypothesize that KpsE's main function is to mediate interactions between the ABC transporter and the outer membrane porin, similar to AcrA/MacA in tripartite efflux pumps. This is now stated in the discussion.

2) It is unclear from the descriptions in the paper how the tripartite complex of KpsMT-KpsE was formed? Did the authors express all three components at the same time in the cells and co-purified them? Or were they separately purified and finally mixed together? The materials and methods do not specify this clearly.

> Thank you for catching that. All three components were co-expressed, which is now clearly stated in the text.

3) Did the authors purify the KpsMT ABC transporter alone and tried to determine its structure? It would have been interesting to see whether KpsE modulates the structure and thus function of KpsMT.

> Yes, indeed. We could not purify WT KpsMT but were successful with the inactivated KpsMT_{E151Q} construct. The transporter was reconstituted into a lipid nanodisc (Msp1e3d1/POPG/POPC) and we determined its structure at about 3.8 Å resolution.

The structure is almost identical to KpsEMT_{E151Q} (RMSD of 1.1Å) and does not provide any additional insights. Therefore, it was not included in the manuscript.

4) The authors show basal ATPase activities of KpsMT-KpsE. Did the authors try to look at stimulated ATPase activity in the presence of the glycolipid? Or by having KpsE present or absent?

> We attempted to measure ATPase activity in the presence of the glycolipid but could not generate reproducible results. This inconsistency is likely due to impurities and batch-to-batch variability of the purified glycolipid. In addition, we noticed significant variations in ATPase activity depending on the type and concentration of solubilizing detergents. Therefore, we consider these measurements as not informative at this time. Efforts to reconstitute KpsMT-KpsE into proper membrane mimetics are underway and will hopefully aid in addressing these points.

As stated above, purification of WT KpsMT was not possible due to dissociation of KpsM and KpsT. Therefore, we currently cannot compare hydrolytic activities in the presence and absence of KpsE in a purified system.

5) The authors used a cross-linked mutant of KspE in functional and structural experiments. I was wondering if they performed *in vivo* experiments (fluorescent microscopy) with WT KspE and noticed any differences with the cross-linked mutant in terms of the speed and extent of hyaluronan (HA) export.

> All functional experiments, including those involving spheroplasts, were conducted with the wild type (non-crosslinked) *P. multocida* KpsE, unless otherwise stated. Only the surface labeling shown in Fig. 1e (two last columns) was generated using the potentially disulfide linked *S. thermodepolymerans* KpsMT-E system. At this point, we don't know whether disulfide stabilization of the KpsE cage affects CPS secretion.

6) The authors mention that in all cryo-EM structures of KpsMT-KspE, the KspE cage looks the same. One may argue that identical conformations were observed because KspE was cross-linked?

> Perhaps. However, the KpsE cage is stabilized by one single disulfide bridge on the periplasmic side. The most dramatic conformational changes of KpsMT occur on the cytosolic side to enable NBD opening and closing. Therefore, any functionally important KpsE transitions in this region could still occur. In addition, non-reducing SDS-PAGE of the purified complex shows monomeric KpsE subunits (alongside high molecular weight cross-linked species), suggesting that not all protomers are oxidized. The observation that KpsMT is catalytically active in the absence of KpsE (see above) supports a model of a static KpsE cage that mediates interactions with KpsD.

7) The authors observe additional cryo-EM densities in protein sample where they added purified substrate and interpret them as glycolipids. Can the authors rule out that this non-proteinaceous density actually represents a molecule of the detergent LMNG used for protein purification? The shape of LMNG resembles the shape of a lipid, at least at the resolution of the cryo-EM density in this region and highly ordered LMNG binding sites are not uncommon.

> We believe some of our maps resolve ordered LMNG molecules at other locations. The solubilization and purification protocol was the same for all samples. Yet, additional densities inside the canyon are only observed after adding the purified glycolipid (ED Fig. 7c and d).

We consider it unlikely that the additional densities observed in the polysaccharide canyon represent LMNG for the following reasons:

- 1) At low contour levels the assigned glycolipid density extends towards the KpsE cage in a tubular shape. This density likely represents the attached polysaccharide chain (see Fig 4c and f). The density is too large to accommodate an LMNG molecule.
- 2) We did not observe similar densities in samples that were not supplemented with the glycolipid (ED Fig. 7c and d).
- 3) When compared, the observed LMNG densities are distinguishable from the glycolipid densities at the resolution obtained (see below).

Reviewer Reports on the First Revision:

Referees' comments:

Referee #1 (Remarks to the Author):

Th authors have address all of my comments!

Referee #2 (Remarks to the Author):

This revised manuscript presents additional data obtained using MINFLUX nanoscopy to visualize capsular polysaccharides, ABC transporter components such as KpsD and lipopolysaccharide. The new data complement and strengthen the conclusions of the work. In particular, the new 2-color data in 3D presented in two new movies are state of the art and show the spatial arrangement of KpsD clusters compared to capsular polysaccharides. Useful explanations on the current limitations of the labeling strategies used, the biological variability and how these could be overcome complete the picture on the whole-cell organization of this ABC transporter and its substrates. All my suggestions and questions were answered satisfactorily.

Referee #3 (Remarks to the Author):

The answers provided by the authors make perfect sense and clarified my open points.

I appreciate that the authors have added an additional spheroplast experiment in Fig. 1g showing that the machinery works also without KpsE under the experimental conditions tested.

I feel that the two additional Figures provided in the rebuttal letter, namely the "naked" structure of KpsMT-EtoQ versus the complex with KpsE, as well as the more in-depth side-by-side comparison of Glycolipid 1 versus LMNG densities have a value for the paper. These information are of direct importance in the context of the main story line (i.e. the potential role of KpsE which is still to some extent elusive + support of a main claim, namely the molecular basis of glycolipid recognition) and should in my view be included as part of the Supplementary Material with a short mentioning of it in the main text.

I do not think that would render the "final product" overly complicated and can be done with reasonable effort.

Author Rebuttals to First Revision:

We would like to thank the referees once again for reviewing our manuscript. We appreciate their feedback.

Referee #1 (Remarks to the Author):

Th authors have address all of my comments!

>> Thank you.

Referee #2 (Remarks to the Author):

This revised manuscript presents additional data obtained using MINFLUX nanoscopy to visualize capsular polysaccharides, ABC transporter components such as KpsD and lipopolysaccharide. The new data complement and strengthen the conclusions of the work. In particular, the new 2-color data in 3D presented in two new movies are state of the art and show the spatial arrangement of KpsD clusters compared to capsular polysaccharides. Useful explanations on the current limitations of the labeling strategies used, the biological variability and how these could be overcome complete the picture on the whole-cell organization of this ABC transporter and its substrates. All my suggestions and questions were answered satisfactorily.

>> Thank you.

Referee #3 (Remarks to the Author):

The answers provided by the authors make perfect sense and clarified my open points. I appreciate that the authors have added an additional spheroplast experiment in Fig. 1g showing that the machinery works also without KpsE under the experimental conditions tested.

>> Thank you.

I feel that the two additional Figures provided in the rebuttal letter, namely the "naked" structure of KpsMT-EtoQ versus the complex with KpsE, as well as the more in-depth side-by-side comparison of Glycolipid 1 versus LMNG densities have a value for the paper. These information are of direct importance in the context of the main story line (i.e. the potential role of KpsE which is still to some extent elusive + support of a main claim, namely the molecular basis of glycolipid recognition) and should in my view be included as part of the Supplementary Material with a short mentioning of it in the main text.

I do not think that would render the "final product" overly complicated and can be done with reasonable effort.

>> The requested information has been included as Extended Data Figures 2j and 7e and is discussed in the text.